

# Shifting cultivation and hunting across the savanna-forest mosaic in the Gran Sabana, Venezuela: facing changes

Izabela Stachowicz[1,2], José R. Ferrer-Paris[3,4] and Ada Sanchez-Mercado[4,5]

[1] Department of Biodiversity Studies and Bioeducation, University of Lodz, Łódź, Poland
[2] Laboratorio de Biología de Organismos, Centro de Ecología, Instituto Venezolano de Investigaciones Científicas, Caracas, Venezuela
[3] Laboratorio de Ecología Espacial, Centro de Estudios Botánicos y Agroforestales, Instituto Venezolano de Investigaciones Cientificas, Maracaibo, Venezuela
[4] School of Biological, Earth and Environmental Sciences, University of New South Wales, NSW, Kensington, Australia
[5] Ciencias Ambientales, Universidad Espíritu Santo, Guayaquil, Samborondón 092301, Ecuador

## ABSTRACT

**Background**. Human encroachment and overexploitation of natural resources in the Neotropics is constantly increasing. Indigenous communities all across the Amazon, are trapped between a population rise and a hot debate about the sustainability of hunting rates. The Garden Hunting hypothesis states that shifting cultivation schemes (conucos) used by Amazon indigenous communities may generate favorable conditions, increasing abundance of small and medium wildlife species close to the 'gardens' providing game for indigenous hunters.

**Methods**. Here, we combined camera trap surveys and spatially explicit interview dataset on Pemón indigenous hunting scope and occurrence in a mosaic of savanna and forest in the Gran Sabana, Venezuela to evaluate to what extent the wildlife resource use corresponds to Garden Hunting hypothesis. We applied the Royle–Nichols model and binomial regression in order to: (1) assess whether abundance of small and medium wildlife species is higher close to conucos and (2) evaluate whether hunters select hunting localities based on accessibility to wildlife resources (closeness to conuco) more than wildlife abundance.

**Results**. We find mixed evidence supporting the Garden Hunting hypothesis predictions. Abundance of small and medium species was high close to conucos but the pattern was not statistically significant for most of them. Pemón seem to hunt in locations dominated by forest, where species abundance was predicted to be higher, than in close vicinity to conucos. Hunting scope was focused on the most abundant species located close to the conuco (*Cuniculus paca*), but also in less abundant and unavailable species (*Crax alector, Tapirus terrestris* and *Odocoileus virginianus*).

**Conclusions**. Our research provided the first attempt of a systematic sampling survey in the Gran Sabana, generating a quantitative dataset that not only describes the current pattern of wildlife abundance, but sets the base-line to monitor temporal and spatial change in this region of highland Amazon. We discuss the applicability of the estimates generated as a baseline as well as, environmental challenges imposed by economic, social and cultural changes such as mining encroachment for wildlife management.

Corresponding author
Izabela Stachowicz,
izabela.stachowicz@biol.uni.lodz.pl

# INTRODUCTION

Biodiversity loss has fueled a vigorous debate about sustainability of the current hunting rates in the Neotropics and particularly in the Amazon basin (*Robinson & Bennett, 2004*; *Lewis, Edwards & Galbraith, 2015*; *Ripple et al., 2016*; *Benítez-López et al., 2017*; *Benítez-López et al., 2019*). Hunting by inhabitants of tropical forests has increased in recent years (*Fa, Peres & Meeuwig, 2002*) due to human population growth, easier access to undisturbed forests, change in hunting technology, scarcity of alternative protein sources, and higher demand for bushmeat (*Bennett & Robinson, 2000*; *Benítez-López et al., 2017*). Worldwide, more than half of the intact forests and wilderness areas are partially devoid of large mammals and birds, with a significant reduction in abundance (*Benítez-López et al., 2017*; *Benítez-López et al., 2019*). However, current estimates of wildlife abundance reduction do not take into account cultural factors such as taboos, religion, traditional hunting technology, and prey preferences that can have a major influence in patterns of resource use in indigenous communities (*Vetter et al., 2011*; *Carvalho et al., 2015*; *Gray, Bozigar & Bilsborrow, 2015*), which have a narrower hunting scope and magnitude compared with non-indigenous hunters (*Antunes et al., 2019*).

Biodiversity patterns in the Amazon have been altered by human societies since pre-Columbian times (*Etter, McAlpine & Possingham, 2008*; *Levis et al., 2017*), but the current rate of transformation and loss are unprecedented and expected to increase in the future (*Lewis, Edwards & Galbraith, 2015*; *Jedrzejewski et al., 2017*; *Curtis et al., 2018*; *Ferrer-Paris et al., 2019*). Indigenous people in the Neotropics typically create forest-agricultural mosaics based on shifting cultivation systems through clearing of small forest plots or "conucos" by slash-and-burn practice (*Warner, 1991*). The Garden Hunting hypothesis (*Linares, 1976*; *Naughton-Treves et al., 2003*) states that modification of plant community and in situ care of domesticated plants in a shifting cultivation scheme, may generate favorable conditions (e.g., high-nutrient, low-toxicity crops and the abundant browse of regenerating vegetation), for adaptable, fast-reproducing species, such as rodents, peccaries, and armadillos (*Constantino, 2019*), but in turn could act as population sink for large carnivores who are systematically hunted when they venture close to the gardens (*Naughton-Treves, 2002*). Interaction between physical and cultural contexts influences the relationship with wild life species, either as a source of protein, pest (*Smith, 2005*) or pets (*Naughton-Treves, 2002*). These local effects cascade across the landscape, ultimately shape regional patterns of wildlife abundance and species diversity that might range in effect from mild declines to more severe cases of "empty forests" (*Redford, 1992*; *Naughton-Treves et al., 2003*; *Smith, 2005*; *Constantino, 2015*; *Bogoni, Peres & Ferraz, 2020*).

The Amazon basin still looks like an exceptional large region of intact forest, but in fact, there are large regional differences in biodiversity patterns, cultural diversity and pressures on natural resources (*Naughton-Treves, 2002*; *ter Steege et al., 2020*). For example, the

Guiana Shield or highland Amazon has lower aboveground live biomass (*Saatchi et al., 2007*), anomalous savanna vegetation and forest-savanna mosaic (*Rull et al., 2013*), nutrient deficiency and low water retention capacity in soils than lowland Amazon (*Dezzeo et al., 2004*). Fauna and flora in highland Amazon show high diversity and endemism (*Huber, Febres & Arnal, 2001*), and lower prevalence of domesticated plants (*Levis et al., 2017*). Low human population density and limited agricultural potential of the lands in highland Amazon, have prevented high rates of land cover change and infrastructure development, and relative lower levels of threats (*Rull et al., 2013*; *Ferrer-Paris et al., 2019*). While in lowland Amazon the role and magnitude of external factors driving increasing hunting rates have been studied on local and regional scale (*Peres, 2000*; *Zapata-Ríos, Urgilés & Suárez, 2009*; *Constantino, 2015*; *Gray, Bozigar & Bilsborrow, 2015*), these patterns remain understudied in the northern, highland Amazon. This is particularly critical since prey abundance or density patterns in this region are poorly known (*Hollowell & Reynolds, 2005*; *Lim et al., 2005*; *Stachowicz et al., 2020*). Indigenous communities in Latin and Central America obtain dietary protein mainly through fishing and hunting (*Bennett & Robinson, 2000*), while shifting cultivation provides them with vegetables, tubers, and some fruits (*Rodríguez, 2004*; *Smith, 2005*). Both activities, shifting cultivation and hunting raise concerns about the sustainability harvest of natural resources, especially because a transparent, legal framework for hunting is missing all across the Amazon region (*Van Vliet et al., 2019*).

Our study focused on Pemón indigenous communities, inhabiting a mosaic of savanna and forest of the Gran Sabana in South Eastern Venezuela, highland Amazon. Extensive agriculture and cattle raising activities are not viable in the Gran Sabana due to the scarcity of nutrients in the soil (*Rodríguez, 2004*; *Rull et al., 2013*). Instead, Pemón indigenous communities practice shifting cultivation, fishing and hunting (*Coppens & Perera, 2008*). They have cultural taboos prohibiting hunting of certain wildlife (e.g., anteaters, foxes, armadillos, sloths, monkeys, and felids such as jaguars and pumas) and preferences for hunting tapirs, deers, peccaries, pacas, turtles and agoutis (*Coppens & Perera, 2008*). Additionally, new religion restriction has emerged recently colliding with traditional customs (*Knoop et al., 2020*).

Here, we combine wildlife occurrence data from the first comprehensive camera-trap survey in the Gran Sabana, and spatially explicit hunting information based on interviews with indigenous communities in order to: (1) describe the Pemón's hunting practice, including scope, occurrence and hunting technology, and (2) evaluate the influence of conucos on animal abundance while controlling for the influence of habitat. Particularly we wanted to test two predictions of the Garden Hunting hypothesis: (a) abundance of small and medium wildlife species is higher close to conucos, and (b) hunters select hunting localities based on accessibility to wildlife resource (closeness to conuco) more than wildlife abundance. To test the first prediction, we fitted occupancy models (*Royle & Nichols, 2003*) to predict relative abundance of medium and small wildlife. For the second prediction, we related localities reported with and without hunting by the interviewees, with variables explaining wildlife abundance and distance to nearest conuco. We further compared the predicted abundance of wildlife in hunting and not hunting sites. This study is intended as

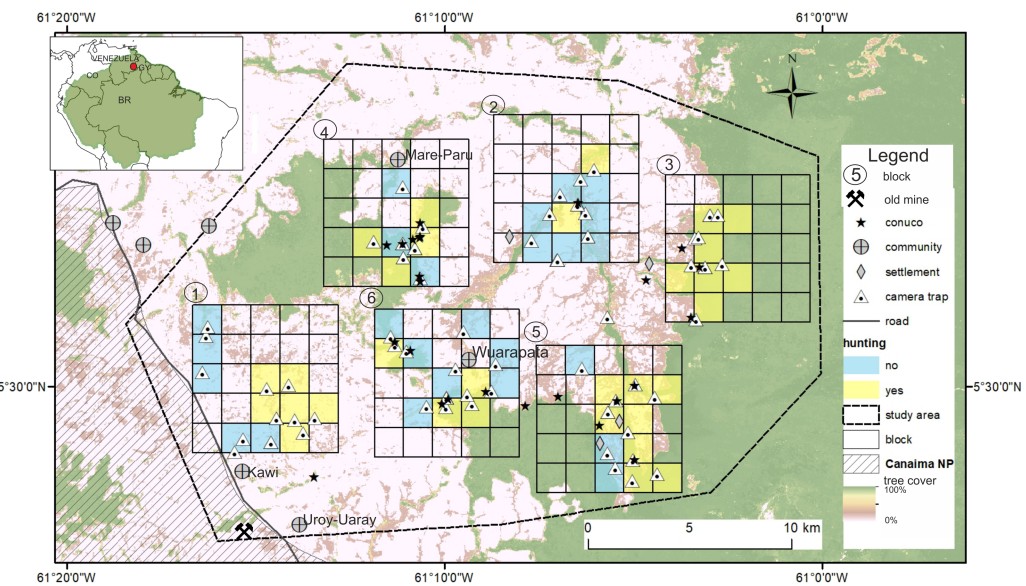

**Figure 1** Study area in the Gran Sabana, Venezuela showing location of the six blocks surveyed with camera traps and the location of conuco.

a baseline evaluation of wildlife presence under human activity in a savanna-forest mosaic in highland Amazon. Although, our recommendations are specific to our case study, our approach to combine different sources of hunting data and species diversity may be widely applied in other regions (*Huang et al., 2020*).

## MATERIALS & METHODS

### Study area

The study area covers 615 km$^2$ at the eastern part of the Gran Sabana on the border of the Canaima National Park, with an elevation range 800–1200 m, close to the Venezuela–Guyana international border (Fig. 1). Vegetation is dominated by scrub (*Clusia* spp. and *Gongylolepis* spp.), broadleaf grassland and savannas of *Axonopus* spp. with scatter patches of gallery forest (*Huber, Febres & Arnal, 2001*). The Ilú and Tramén tepuis massif are surrounded by continuous evergreen montane forest. Average temperatures are between 18 and 24 °C and total annual rainfall is 2000–3000 mm with a dry season (<60 mm / month) from December to March (*Rull et al., 2013*). The Pemón are the only indigenous people inhabiting the Gran Sabana. There are four communities within the study area: Kawi (1100 m; - 61.243W; 5.451N; 50 people 2016), Mare-Paru (884 m; - 61.184W; 5.594N; 45 people in 2016), Uroy-Uaray (1,093 m; - 61.232 W; 5.442 N; 150 people in 2016) and Wuarapata (896 m; - 61.157; W 5.512N; 50 people in 2016; information about the number of inhabitants was obtained from community leaders or *capitanes*).

### Hunting activity

We used a direct, semi-structured interview approach to get information about hunting and conuco occurrence within the six *blocks* (see *Sampling design and camera trap*

*survey* section) (*Carvalho et al., 2015*). We used snowball sampling to identify interview participants. Snowball sampling uses existing study subjects to recruit future subjects from among their acquaintances (*Voicu, 2011*). We initially identified and contacted five community leaders. These then contacted and recruited local hunters and farmers, and so on, until we identified 29 people that were willing to be interviewed: three women and 26 men, all > 18 years old. All were indigenous from Wuarapata (11 people), Uroy-Uaray (8), Kawi (5) and Mare-Paru (5) communities. Interviewees represented 10% of the total population size and were representative in terms of age distribution (mean 44 years old; 22–70 years old). The gender unbalance in the sample likely reflects the role of male as spokesperson in their family group (*Coppens & Perera, 2008*).

All communities represented by their authority –*capitan* –agreed to participate in the research and interview survey, as required by Venezuelan indigenous legislation (La Asamblea Nacional de la Republica Bolivariana de Venezuela, 2005). We obtained verbal informed consent from each participant, after explaining research objectives and assuring participants that information would be used only for research and presented in aggregate analyses, protecting each participant's identity by assigning a numeric code to anonymize participants (*Buppert & McKeehan, 2013*). There was no compensation for participation. The questionnaire and protocol were approved by Dr. Stanford Zent from the Human Ecology Laboratory of the Venezuelan Institute of Scientific Research (July 2015), who acted as external ethical committee.

An interviewee was considered reliable if a participant could differentiate regional from not-regional animals (e.g., *Tremarctos ornatus*) shown in pictures and drawings (plates of *Linares, 1998*) and if the person has been living in the community on the Gran Sabana for most of his/her life. We interviewed each participant independently to minimize biased responses (*Jones et al., 2008*). We conducted the interviews in Spanish, using a local translator of Arekuna Pemón's dialect when required, and registering the species' local name in Arekuna Pemón's dialect.

We assumed that hunting trip was the main hunting method used (*Urbina, 1979*), and our interview survey focused on obtaining baseline information about three aspects of hunting trip activity: (1) hunting occurrence; (2) hunting scope (which species are most important in term of perceived value and preference); and (3) hunting technologies. We specifically asked about the following topics: (1) whether they currently hunt or not in the vicinity of the conuco and whether they did in the past; (2) the list of hunted species, both mammals and birds; (3) the three most preferred hunted species, being the first species the most preferred; (4) preferred hunting areas; (5) preferred hunting season; (6) occupation (mining, tourism, etc.); (7) food sources (conuco, fishing and hunting, processed food); and (8) hunting technology used on hunting trips. Besides direct questioning, we also evaluated hunting technology by reviewing the pictures from the camera trap survey (see next section) looking for evidence of hunters, hunters with firearms or dogs. In each sampling unit where the camera trap was installed, we asked whether they hunt (1) or not (0) to obtain spatial distribution of hunting occurrence in the study area. To accurately identify animals hunted, and avoid misinterpretation with animals' local names, we showed pictures and illustrations of wildlife (*Linares, 1998*) to the participants.

To identify which species are important game species for Pemón people, we used two criteria: the frequency a given species is reported as target game, and how frequently it is mentioned as preferred game. For that we calculate two indexes for each species, importance of hunting ($Hv$) and hunting preference ($Pv$) (*Carvalho et al., 2015*). Both indexes correct the bias introduced by sampling size in the species citation rate, by multiplying the number of informants giving information on each species. $Hv$ is defined as:

$$Hv = \sum \left( \frac{h}{n} \right) xN, \tag{1}$$

where $h$ is the number of times a species is mentioned as a targeted animal, $n$ the total number of citations for all species, and $N$ the number of interviewees (*Carvalho et al., 2015*, modified from *Phillips et al., 1994*; *Fernandes-Ferreira et al., 2012*).

The hunting preference index (Pv), measure the frequency that each species is cited as the first option for hunting among others, and is defined by:

$$Pv = \sum \left( \frac{p}{n} \right) xN, \tag{2}$$

where $p$ is the number of times a species is cited as the first option (among the three most preferred hunted species), $n$ the total number of citations for all prey species, and $N$ the number of interviewees. In this case, zero values (i.e., no preference) were excluded.

Due to the nature of the questions, the $Hv$ and $Pv$ should be interpreted as the perceived importance and preference. Since we don't have counts of hunted individuals, we cannot calculate frequency of hunting or consumption per species. We expect that the self-reported species lists reflect those that are of highest valuable for Pemón people, but under some circumstances the species consumed less frequently could be more reliably reported by recalls (more memorable), than taxa that are consumed frequently (*Golden et al., 2013*).

In some cases, Pemón names do not match scientific names and for example, they use "savanna deer" or "forest deer" ambiguously for red brocket (*Mazama americana*) and gray brocket (*Mazama gouazoubira*), and "armadillo" for the greater long-nosed armadillo (*Dasypus kappleri*) and the nine-banded armadillo (*Dasypus novemcinctus*). So in these cases, the values for $Hv$ and $Pv$ were calculated at genus level.

## Sampling design and camera trap survey

We used data from a previous camera trap survey conducted between September 2015 –April 2016. The original sampling design was developed to optimize covering habitat diversity in order to evaluate how mammal species richness is related to habitat types, and is described in detail in *Stachowicz et al. (2020)*, but we provide here a brief summary of the initial setting and how we adapted data *a posteriori* for our analysis. Sampling design comprised six 50 km$^2$ blocks within the study area (B01–B06). Blocks were selected to represent landscapes with different configurations of forest, savanna and shrubs habitats. Since only 30 cameras were available, sampling was divided into three periods of 60-days each, and in each period a two-levels stratified random sampling was used to select 30 sampling units (five in each block) for camera deployment. This stratification ensures a balanced representation of sampling units with different coverage of habitat types and

fragmentation in each block during each period. As a side effect of this, some sampling units with unique values within each block (sites with high tree cover within a block dominated by savanna) were selected for sampling in two or three periods and those cameras were neither relocated nor replaced.

We also recorded direct observations and indirect evidence (scats, tracks, scratches on trees, burrows, etc.) of animal presence along the routes walked during field work, and recorded their coordinates with GPS. We had a total of $n = 159$ records during 29 days of camera deployment and maintenance, with a mean of 16.8 km walked each day.

Total sampling effort was 4,548 cameras per day, which resulted in 771 detection events for mammals and 226 events for birds across 60 camera sites (Table 1). Events represent sequences of photos separated by less than 5 min and showing the same animal species and presumably the same individuals. We identified mammal species (*Eisenberg, 1989*; *Linares, 1998*) and birds (*Hilty, Tudor & Gwynne, 2003*) using reference works for Venezuela and South America. We calculated the index of frequency of detection for all mammals and bird species registered with camera traps as the number of detection events for species per 100 days of camera trapping (*O'Brien et al., 2010*) in order to have available information to compare with similar studies in Latin America.

## Predictive variables: Tree cover, distance to river and distance to conuco

For this study, we calculated tree cover, distance to nearest conuco and river as covariates of species abundance (Supplemental Information 1). Previous analysis suggested that most mammal species in the study area are more associated with forest habitat than shrubs or savanna (*Stachowicz et al., 2020*). We used mean tree cover (in percentage) as a quantitative variable correlated with these habitat types and consistent with metrics used for the sampling design. Mean tree cover was calculated from remote sensing products (*Hansen et al., 2013*) using a 1 km buffer around the camera location "*tree_buffer*". This variable has a bimodal distribution with a lower mode at 10–20% corresponding with the savanna, a higher mode at 70–80% corresponding with forest, and intermediate values roughly corresponding with the less common shrub habitat. We used 1 km radius under the assumption that it is wide enough to represent the area of the most abundant game species home range (lowland paca *Cuniculus paca* 2–3 ha, *Jorge & Peres, 2005*; red-rumped agouti *Dasyprocta leporina* 3.4–1.6 ha, *Benavides, Arce & Pacheco, 2017*) and narrow enough to maintain variability in tree cover within the scale of a camera trapping site (*Scotson et al., 2017*).

Presence of rivers is considered an important variable explaining temporal and geographic patterns of distribution and abundance of mammals (*Constantino, 2015*; *Hedwig et al., 2018*). During the dry season when the camera traps were placed, access to water is an important resource. We calculated distance to nearest rivers (in meters) using vector data of rivers (*Señaris, Lew & Lasso, 2009*).

During fieldwork we marked with a GPS the location of active and recently abandoned *conucos* ($n = 25$) identified in situ, and hunting sites ($n = 32$) reported by interviewees and confirmed by the local guides (Fig. 1). Distance from each camera to the nearest conuco

**Table 1  Nonvolant medium, large mammal and bird species detected during the camera trap survey.**

| Common name | Scientific name | Species' name in Arekuna | Frequency of detection (FD) | Number of detection events (D) | Survey method | IUCN Red List Category | Dietary group |
|---|---|---|---|---|---|---|---|
| **ARTIODACTYLA** | | | | | | | |
| Red Brocket | *Mazama americana* (Erxleben 1777) | kutsari | 0.44 | 20 | CT,TRK, INT | DD | herb |
| Gray Brocket | *Mazama gouazoubira* (G.Fischer 1814) | kariyawku | 1.03 | 47 | CT,TRK, INT | LC | herb |
| White-tailed Deer | *Odocoileus virginianus* (Zimmermann, 1780) | waikín | 0.09 | 4 | CT,TRK, INT | LC | herb |
| Collared Peccary | *Pecari tajacu* (Linnaeus 1758) | poyinke | 0.04 | 2 | CT,TRK, INT | LC | omni |
| White-lipped Peccary | *Tayassu pecari* (Link 1795) | pakirá | 0.04 | 2 | CT, INT | VU | omni |
| **CARNIVORA** | | | | | | | |
| Margay | *Leopardus wiedii* (Schinz 1821) | – | 0.04 | 2 | CT | VU | carn |
| Ocelot | *Leopardus pardalis* (Linnaeus 1758) | kaukan | 0.33 | 15 | CT,TRK, INT | LC | carn |
| Jaguar | *Panthera onca* (Linnaeus 1758) | temenen | 0.31 | 14 | CT,TRK, INT | VU | carn |
| Puma | *Puma concolor* (Linnaeus 1771) | kusariwara | 0.24 | 11 | CT,TRK, INT | LC | carn |
| Crab-eating Fox | *Cerdocyon thous* (Linnaeus, 1766) | maikan | 0.97 | 44 | CT,TRK, INT | LC | omni |
| Tayra | *Eira barbara* (Linnaeus 1758) | yeruena | 0.46 | 21 | CT, INT | LC | carn |
| South American Coati | *Nasua nasua* (Linnaeus 1766) | kuachi | 0.18 | 8 | CT, INT | LC | omni |
| **CINGULATA** | | | | | | | |
| Greater Long-nosed Armadillo | *Dasypus kappleri* (Krauss 1862) | – | 0.75 | 34 | CT, TRK | LC | inse |
| Nine-banded Armadillo | *Dasypus novemcinctus* (Linnaeus 1758) | muruk | 0.42 | 19 | CT,TRK, INT | LC | inse |
| Southern Naked-Tailed Armadillo | *Cabassous unicinctus* (Linnaeus 1758) | – | 0.04 | 2 | CT | LC | inse |
| Giant Armadillo | *Priodontes maximus* (Kerr 1792) | mauraimu | 0.18 | 8 | CT,TRK, INT | EN | inse |
| **PERISSODACYLA** | | | | | | | |
| South American Tapir | *Tapirus terrestris* (Linnaeus 1758) | maikuri | 0.31 | 14 | CT,TRK, INT | VU | herb |
| **DIDEPHIMORPHIA** | | | | | | | |
| Guianan White-eared Opossum | *Didelphis imperfecta* (Mondolfi and Pérez-Hernandez 1984) | – | 0.31 | 14 | CT | LC | omni |
| Common Opossum | *Didelphis marsupialis* (Linnaeus 1758) | awaré | 0.04 | 2 | CT, INT | LC | omni |
| **PILOSA** | | | | | | | |
| Southern Tamandua | *Tamandua tetradactyla* (Linnaeus 1758) | woiwo | 0.13 | 6 | CT, INT | LC | omni |
| Giant Anteater | *Myrmecophaga tridactyla* (Linnaeus 1758 | wareme | 0.33 | 15 | CT,TRK, INT | VU | insec |
| **RODENTIA** | | | | | | | |
| Lowland Paca | *Cuniculus paca* (Linnaeus 1766) | uraná | 5.83 | 265 | CT,TRK, INT | LC | herb |
| Red-rumped Agouti | *Dasyprocta leporina* (Linnaeus 1758) | akuri | 4.2 | 191 | CT,TRK, INT | LC | herb |
| Capybara | *Hydrochoeris hydrochaeris* (Linnaeus 1766) | parwena | 0.07 | 3 | CT,TRK, INT | LC | herb |
| **PRIMATES** | | | | | | | |
| Wedge-capped Capuchin | *Cebus olivaceus* (Schomburgk, 1848) | ibarakao | 0.18 | 8 | CT,TRK, INT | LC | omniv |
| Guyanan Red Howler | *Alouatta macconnelli** (Linnaeus 1766) | arauta | – | – | TRK, INT | LC | herb |

**Table 1** (*continued*)

| Common name | Scientific name | Species' name in Arekuna | Frequency of detection (FD) | Number of detection events (D) | Survey method | IUCN Red List Category | Dietary group |
|---|---|---|---|---|---|---|---|
| | | | | **BIRDS** | | | |
| Pectoral sparrow | *Arremon taciturnus* | | 0.18 | 8 | **CT, INT** | **LC** | |
| Savanna hawk | *Buteogallus meridionalis* | woroiwo | 0.02 | 1 | **CT, TRK, INT** | **LC** | |
| Turkey vulture | *Cathartes aura* | kurüm | 0.04 | 2 | **CT, TRK, INT** | **LC** | |
| Black curassow | *Crax alector* | pauwi | 1.06 | 48 | **CT, TRK, INT** | **VU** | |
| Tinamous | *Crypturellus spp.* | | 0.04 | 2 | **CT, TRK, INT** | | |
| Variegated tinamou | *Crypturellus variegatus* | | 0.15 | 7 | **CT** | **LC** | |
| Little tinamou | *Crypturellus soui* | churima | 0.24 | 11 | **CT, INT** | **LC** | |
| Ruddy quail-dove | *Geotrygon montana* | | 0.02 | 1 | **CT** | **LC** | |
| Grey-fronted dove | *Leptotila rufaxilla* | wakuma | 1.5 | 68 | **CT, TRK, INT** | **LC** | |
| Green ibis | *Mesembrinibis cayennensis* | | 0.02 | 1 | **CT, TRK, INT** | **LC** | |
| Tropical mockingbird | *Mimus gilvus* | paraura | 0.24 | 11 | **CT** | **LC** | |
| Rufous-winged ground cuckoo | *Neomorphus rufipennis* | | 0.02 | 1 | **CT** | **LC** | |
| Spix's guan | *Penelope jacquacu* | wora | 0.18 | 8 | **CT, TRK, INT** | **LC** | |
| Great tinamou | *Tinamus major* | marú | 0.53 | 24 | **CT, TRK, INT** | **NT** | |
| White-necked thrush | *Turdus albicollis* | | 0.73 | 33 | **CT** | **LC** | |

**Notes.**

Scientific, common names and names in Arekuna (Pemn dialect) are provided for each species. Frequency of detection index, total number of detections for species and survey method: are shown.

CT, camera trapping; TRK, tracking; INT, interviews with local Pemn communities.

IUCN Red List category based on country assessment (*Rodríguez, Garcia-Rawlins & Rojas-Suárez, 2015*) and dietary group based on *Linares (1998)* and *Eisenberg (1989)* are shown for mammals (herb herbivorous, omni omnivorous, carn carnivorous and inse insectivorous).

*Species documented only by vocalization and interviews with local communities.

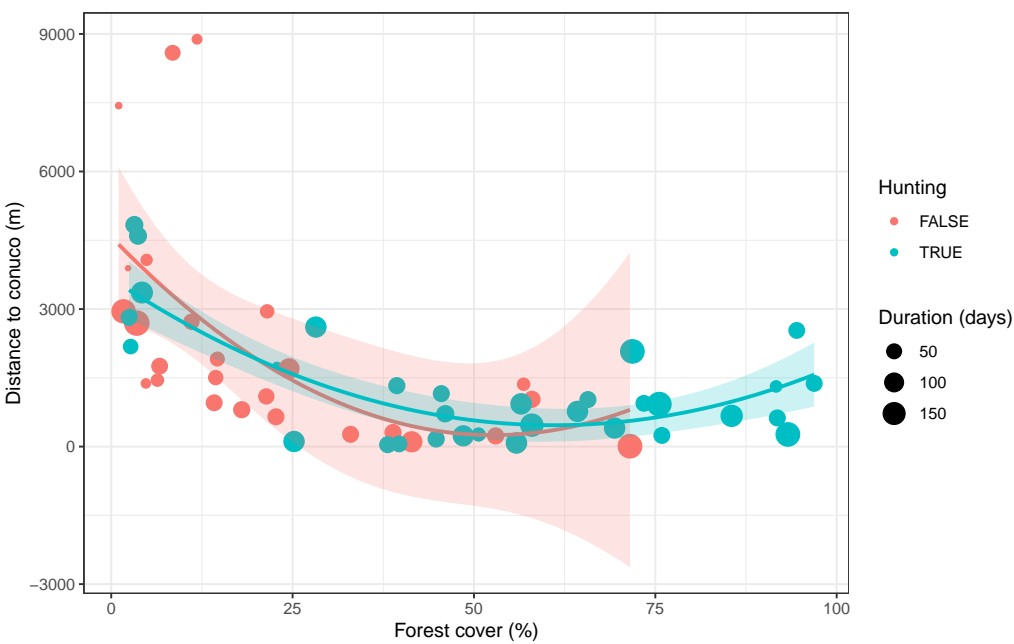

**Figure 2** **Distribution of predictive variables used to test the Garden Hunting hypothesis.** Values of the percentage of tree cover, distance to conuco and hunting occurrence across sampling units are shown.

was calculated using the GPS coordinates from cameras and conucos. This variable had an asymmetric distribution with a mean value of 1.58 km and a range from 0 to 8 km. We also recorded which cameras were located adjacent or near reported hunting sites (binomial variable *hunting*, FALSE $n = 23$, TRUE $n = 34$).

After visual inspection of the distribution of tree cover, hunting occurrence, and distance to conuco, we decided to discard three cameras with extreme distance values. We also discarded four cameras that were active for less than seven days. Thus all following analyses count on data from 54 cameras within 5 km of the nearest conuco, with more than seven days of activity (Fig. 2).

## Data analysis: testing Garding Hunting predictions

To evaluate the prediction of Garden Hunting Hypothesis we followed a three-step approach using occupancy model, Chi-square test and logistic regression.

To test the prediction whether abundance of small and medium wildlife species is higher close to conucos, we need a measure of influence of conucos while controlling for the influence of habitat on species abundance, and the spatial and temporal heterogeneity in probability of detection. For that, we fit a hierarchical Bernoulli/Poisson $N$-mixture model (*Royle & Nichols, 2003*) for each species to evaluate how the probability of occupancy relates to tree cover and distance to conuco, allowing for abundance-induced heterogeneity in detection probability. These models are a type of latent abundance mixture models, and are often referred as Royle–Nichols models, or RN-models. They are based on the assumption that the detection probability at a survey point $p_{ij}$ depends on the species' site-specific

abundance $N_i$:

$$p_{ij} = 1 - (1 - r_{ij})^{Ni}, \tag{3}$$

where $r_{ij}$ is the detection probability of a single individual. Repeated visits at a survey point generate a history of detection/non-detection events $y_{ij}$, from which $p_{ij}$ is estimated. The abundance state ($N_i$) of site $i$ was modeled as $N_i \sim$ Poison ($\lambda_i$), while the observation process was modeled as $y_{ij} / N_i \sim$ Bernoulli ($p_{ij}$). Estimation of $p_{ij}$ allows us to draw conclusions about $N_i$.

In order to build detection histories for species recorded during the camera trap survey, we considered each camera location as a "site" ($i$; 54 in total). We divided the total sampling period of 180 days into several "visits" ($j$). We tested different values of visit duration and found that a duration of 21 days reduced the prevalence of zeroes but maintained enough number of visits (three for each sampling period, up to nine for cameras fixed in the same location) to successfully fit the model.

Covariates of $N_i$ (site covariates) and $p_{ij}$ (observation covariates) were modeled using the logit link. We explored several covariates with alternative parameterizations to ensure best possible model fit given the restricted sample size and low number of detection for some species (see Supplemental Information 1 for details). As site covariates we used tree cover percentage around the camera trap (*tree_buffer*), distance to river (*dist_river*), and distance to nearest conuco (*dist_conuco*), all variables were standardized to zero mean and unit standard deviation. Since a couple of species might be associated with shrub habitat with intermediate values of tree cover (*Stachowicz et al., 2020*) we added a quadratic term (*tree_buffer*$^2$) to their models.

We used sampling date, sampling effort (camera/day), and density of direct and indirect off-camera records to account for spatial and temporal heterogeneity in detectability (*Cubaynes et al., 2010*). Sampling date (*date*) was recorded as the number of days since the start of sampling (21 September 2015) to the beginning of the "visit" and standardized to zero mean and unit standard deviation. Sampling effort (*effort*) was calculated as the number of days the camera remained active divided by the duration of the visit. Thus, *effort* was always $\leq 1$, and was set to empty value (*NA*) when the camera was not present or inoperative during the whole duration of the visit. We calculated tracks density (*track_dens*) as the inverse distance weighted sum of wildlife activity recorded off-camera (direct observations and indirect evidence such as tracks, scratches, cavities and excrement of animal presence during field-work):

$$track\_dens_i = \sum_{j=1}^{k} 1/d_{ij}^q, \tag{4}$$

where $d_{ij}$ is the distance between camera $i$ and record $j$ for all $k = 159$ records, and $q$ is a fixed power parameter that influences the degree of smoothing in the interpolation, we used $q = 0.25$ based on visual inspection. The result was standardized to zero mean and unit standard deviation.

For each species, we fitted a full model including the three observation covariates ($p$ (*date + effort + tracks_dens*)) and the three site covariates ($\lambda$ (*tree_buffer + tree_buffer*$^2$ +

*dist_river* + *dist_conuco*)) using the *occuRN* function of the R package unmarked (*Fiske & Chandler, 2011*). We assessed model fit for the full model using goodness of fit test based on Pearson $\chi^2$ and parameter bootstrapping with 10,000 samples, and inspecting under- or overdispersion ($\hat{c}$, calculated by dividing the observed $\chi^2$ statistic by the mean of the statistics obtained from bootstrap samples), magnitude of parameter estimates and standard errors, and predicted values of the state variable at the sample locations (*MacKenzie & Bailey, 2004*; *Royle, 2006*). For species with a suitable full model (with > 10 detections), we proceeded to create a model selection table with all combinations of covariates (32 models for species with linear effect of tree cover and 48 for species with quadratic effect of tree cover), ranked models according to information criteria corrected for small sample size (AICc if $\hat{c} \leq 1$ or QAICc if $\hat{c} > 1$), and the corresponding $\Delta$(Q)AICc and model weights.

We assessed the relative importance of each detection and occupancy covariate by calculating the sum of weights of the model containing that variable (*Burnham & Anderson, 2002*; *Symonds & Moussalli, 2011*). Values range from zero to one indicating increasing levels of support, and we use an informal scale to describe the level of support as very strong (>0.9), strong (0.6– 0.9), moderate (0.3–0.6) and low (<0.3). We further calculated model averaged coefficients and predictions of the state variable ($\lambda$) based on the subset of models with $\Delta$(Q)AICc $\leq 10$ (*Burnham, Anderson & Huyvaert, 2011*; *Mazerolle, 2020*). In case of overdispersion ($\hat{c} > 1$) we assumed the lack of fit is due to unaccounted sources of error and used the value of $\hat{c}$ to inflate the standard errors and confidence intervals. For underdispersed models ($\hat{c} \leq 1$), no modification to standard errors or intervals was made, but consider these as conservative assessments of uncertainty (*Kery & Royle, 2015*).

To evaluate the prediction whether hunters select hunting localities based on accessibility to wildlife resources we first used the interview responses on vegetation type and season with contingency tables, to evaluate which season and habitat type are used as hunting localities. For that, we tabulated the number of interview responses from each community for the three levels of preferred hunting vegetation types (forest, savanna and mixed) and the two levels of hunting seasons (dry, and rainy season). We used the $\chi 2$ (Chi-square) test to assess the significance of the relationship between variables.

Second, we used the data collected during field work at 53 sites with cameras and fitted a logistic regression to the binomial *hunting* variable with formula:

$$\text{logit}(hunting) \sim \beta_0 + \beta_1 tree\_buffer + \beta_2 dist\_conuco + \beta_3 dist\_river \tag{5}$$

Third, we compared the prediction of latent abundance of the RN-models of each species at these 53 sites, and compared values at sites with and without reported hunting .

## Ethical standards

The study received permits from Ministerio del Poder Popular para Ecosocialismo y Aguas 1419/3/33/2015 and Instituto Nacional de Parques (INPARQUES) 18/16 205, 156, 17 in Venezuela, as well as from the indigenous authorities at each community. The instrument and interview protocols used in Pemón communities were approved and widely used by Fundación la Salle in Venezuela.

## RESULTS

### Frequency of detection of mammals and birds

Camera traps detected a total of 25 species of mammals and 15 of birds of which four species were detected once, seven were detected twice and the remaining 29 were detected three times or more (Table 1, Fig. S1). The most frequently detected (FD) with high number of detections events (D) mammals species were the lowland paca (FD = 5.38; D = 265), the red-rumped agouti (FD = 4.2; D = 191), and the gray brocket (FD = 1.03; D = 47). Among birds, the frey-fronted dove (*Leptotila rufaxilla*) (FD = 1.05; D = 68), and the black curassow (*Crax alector*) (FD = 1.06; D = 48) were the most frequently detected (Table 1). The species with the lowest frequency of detection were margay (*Leopardus wiedii)* (FD = 0.04; D = 2), the white-lipped peccary (*Tayassu pecari*) (FD = 0.04; D = 2), collared peccary (*Pecari tajacu*) (FD = 0.04; D = 2), white-tailed deer *(Odocoileus virginianus)* (FD = 0.09; D = 4), Southern naked-tailed armadillo (*Cabassous unicinctus*) (FD = 0.04; D = 2), common opossum (*Didelphis marsupialis*) (FD = 0.04; D = 2), and capybara (*Hydrochoeris hydrochaeris*) (FD = 0.07; D = 3) (Table 1). During the interviews with Pemón the majority of species registered by camera trap were recognized, except margay and Southern naked-tailed armadillo, while giant armadillo (*Priodontes maximus*) was only recognized by older interviewees.

### Hunting practice: scope, occurrence, and hunting technology

Of the 29 participants, 19 described themselves as active hunters, five as inactive hunters who hunted in the past and five were no hunters (including the three women interviewed).

The most frequent food sources were agriculture (79%) and fishing and hunting (65%), followed by consumption of processed food (51) (multiple choice was permitted). Only 14% of interviewees identified hunting as an occupation that they carried out. Among other activities carried out, almost all indicated agriculture (99%) and a large proportion indicated fishing (86%), mining (37%), tourism (34%) and others (27%): handicrafts, raising of livestock and transport (they could choose more than one activity). The majority of interviewees (79%), reported that hunted meat was consumed within the family or the community. There was no evidence of commercial hunting - sale of meat, leather or other products derived from the preys.

The most frequent hunting technology used during hunting trips was the shotgun (79%) (Figs. 3A and 3B), while traditional bow and arrows (6%; Fig. 3C), and sling to hunt the birds (10%; Fig. 3D) have recently gained importance due to limited availability of ammunition, 27%). The use of dogs was not reported by interviewees but hunting dogs were visible in three out of nine events of hunters detected by camera traps, where dogs accompanied armed people (Fig. 3A).

According to interviewees, at least nine species of mammals and three species of birds were important game species for Pemón people (Table 2). We detected all of these species with the camera trap survey (see below). The most important species (the highest *Hv* and P*v* values; Table 2) were the white-tailed deer, lowland paca, and black curassows. Red-rumped agouti and South American tapir (*Tapirus terrestris*) were also hunted, but

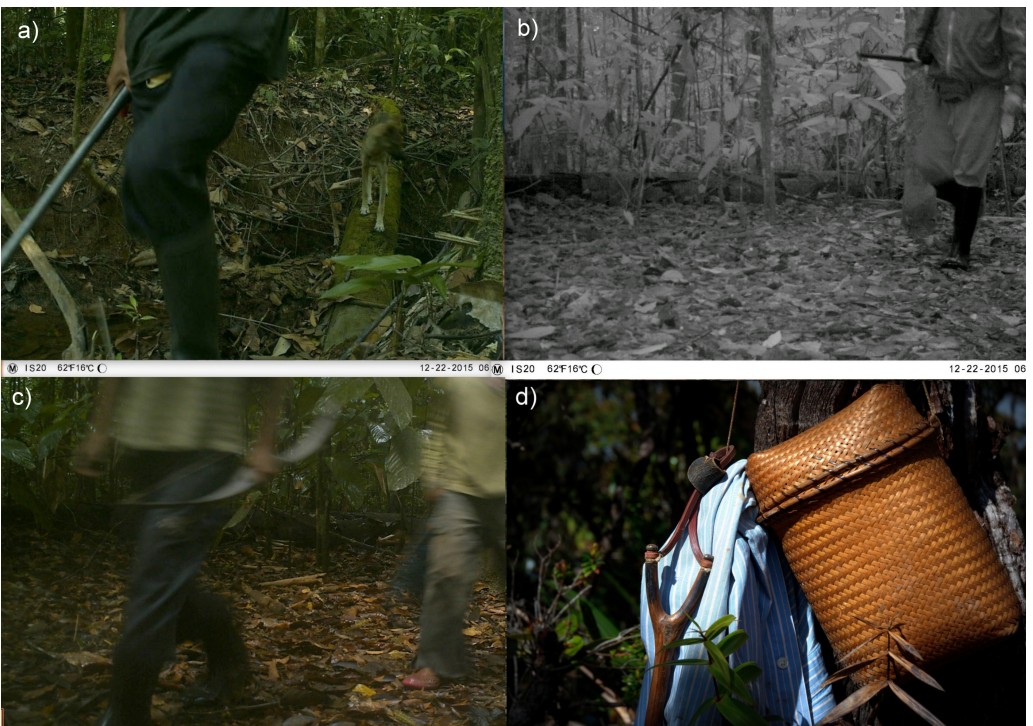

**Figure 3 Hunting technologies used by Pemón.** (A) Hunter with a shotgun and dog captured by camera traps; (B) hunter with a shotgun; (C) hunter with a bow captured by camera traps, (D) sling. Photo credit (D) Izabela Stachowicz.

were not mentioned as preferred game (Table 2). Ten percent of interviewees reported that they only hunt and consume deer meat due to religion restrictions.

## Occupancy models

We explored latent abundance mixture model for 25 mammals and four bird species but discarded models for species with poor fit (Supplemental Information 1 and 2). We completed model selection and averaging for 12 mammal and three bird species with more than ten detections and reasonably good fitting models. Among these 15 species, only ocelot (*Leopardus pardalis*) ($\hat{c} = 1.77$) and nine-banded armadillo ($\hat{c} = 1.29$) showed sign of overdispersion (Table 3), most species showed under-dispersion, with most values between 0.3 and 0.9 except the low value for Great Tinamu (*Tinamus major*) ($\hat{c} = 0.21$).

All variables received some degree of support for all species ($\Sigma$(Q)AICw $> 0.23$; Table 3). Among the covariates of probability of detection, sampling effort had the highest level of support in most species (six species with strong or very strong support, Table 3), except for *D. imperfecta* (*track density* received most support), nine-banded armadillo (*date*), while three species had similar low values for all three covariates (jaguar (*Panthera onca*), ocelot and great tinamu; Table 3).

Among the covariates of lambda, tree cover had strong to very strong support for most of the species except ocelot (moderate), giant anteater (*Myrmecophaga tridactyla*) (moderate) and lowland paca *(low)*. For most species tree cover was modelled as a linear variable,

**Table 2 Indices of hunting importance (Hv) and hunting preference (Pv) reported for the Pemón communities.** Mammal species are ordered by diet groups, birds are presented in one group as they have a mixed diet.

| Diet group | Species | Common name | Hv | Pv | Red List of species |
|---|---|---|---|---|---|
| | | **Mammals** | | | |
| Insectivorous | *Dasypus kappleri* (Krauss, 1862) | greater long-nosed armadillo | 0.244 | | LC |
| | *Dasypus novemcinctus* (Linnaeus, 1758) | nine-banded armadillo | 0.244 | | LC |
| Omnivorous | *Tayassu pecari* (Link, 1795) | white-lipped peccary | 0.975 | | VU |
| Herbivorous | *Tapirus terrestris* (Linnaeus, 1758) | South American tapir | 2.681 | | VU |
| | *Cuniculus paca* (Linnaeus, 1766) | lowland paca | 6.336 | 1.218 | LC |
| | *Dasyprocta leporina* (Linnaeus, 1758) | red-rumped agouti | 2.681 | | LC |
| | *Odocoileus virginianus* (Zimmermann, 1780) | white-tailed deer | 6.823 | 4.874 | LC |
| | *Mazama americana* (Erxleben, 1777), | red brocket | 0.731 | | DD |
| | *Mazama gouazoubira* (G. Fischer, 1814) | gray brocket | 0.731 | | LC |
| | | **Birds** | | | |
| | *Tinamus major* (Gmelin, 1789) | great tinamu | 1.949 | | NT |
| | *Crax alector* (Linnaeus, 1766) | black curassow | 4.630 | 0.975 | VU |
| | *Penelope jacquacu* (Spix, 1825) | spix's guan | 1.949 | | LC |

**Table 3 Model performance metrics.** The *MacKenzie & Bailey (2004)* test on the full model including goodness of fit test based on Pearson ($\chi^2$), estimated dispersion parameter (c-hat) and significant level (*p*). The relative importance of each detection covariate is represented by the sum of AICc or QAICc weights ($\Sigma$AICw) of the model containing that variable. Variables with strong level of support ($\Sigma$AICw > 0.6) are in bold.

| Species | Total detections | *MacKenzie & Bailey (2004)* test on full model | | | Detectability | | | Lambda | | | |
|---|---|---|---|---|---|---|---|---|---|---|---|
| | | $\chi^2$ | *p* | c-hat | *effort* | tracks_dens | *date* | tree_buffer | tree_buffer$^2$ | dist_conuco | dist_river |
| *Dasyprocta leporina* | 66 | 1093.79 | 0.715 | 0.521 | **0.98** | 0.42 | 0.29 | **1.00** | – | 0.31 | 0.43 |
| *Cuniculus paca* | 71 | 966.51 | 0.82 | 0.44 | **0.97** | **0.92** | 0.25 | 0.30 | – | **0.87** | 0.31 |
| *Leptotila rufaxilla* | 33 | 650.08 | 0.63 | 0.36 | **1.00** | 0.22 | 0.37 | 0.85 | 0.79 | 0.31 | 0.23 |
| *Cerdocyon thous* | 22 | 1217.31 | 0.44 | 0.59 | 0.42 | 0.26 | 0.23 | 0.54 | – | 0.31 | 0.31 |
| *Dasypus novemcinctus* | 17 | 956.26 | 0.14 | 1.29 | 0.22 | 0.23 | **0.85** | 0.41 | – | 0.32 | 0.23 |
| *Crax alector* | 31 | 1098.64 | 0.52 | 0.62 | **0.73** | **0.71** | 0.24 | **0.98** | – | **0.64** | 0.23 |
| *Leopardus pardalis* | 14 | 1427.13 | 0.13 | **1.77** | 0.24 | 0.25 | 0.23 | 0.35 | – | 0.26 | 0.26 |
| *Panthera onca* | 12 | 427.28 | 0.35 | 0.85 | 0.23 | 0.26 | 0.24 | 0.68 | – | 0.26 | **0.68** |
| *Dasypus kappleri* | 25 | 922.37 | 0.47 | 0.76 | 0.65 | 0.49 | 0.46 | **1.00** | – | 0.47 | 0.45 |
| *Mazama gouazoubira* | 33 | 846.97 | 0.65 | 0.52 | **0.97** | 0.22 | 0.22 | **1.00** | – | 0.57 | 0.30 |
| *Didelphis imperfecta* | 11 | 292.12 | 0.42 | 0.57 | 0.45 | **0.96** | 0.23 | 0.38 | – | 0.24 | 0.24 |
| *Tinamus major* | 18 | 319.06 | 0.91 | 0.21 | 0.23 | 0.29 | 0.24 | **0.97** | – | 0.25 | 0.24 |
| *Mazama americana* | 17 | 242.79 | 0.76 | 0.32 | **1.00** | 0.23 | **0.88** | **0.98** | – | 0.24 | 0.25 |
| *Myrmecophaga tridactyla* | 13 | 413.17 | 0.32 | 0.83 | 0.48 | 0.60 | 0.23 | 0.35 | – | 0.38 | **0.89** |
| *Eira barbara* | 16 | 282.14 | 0.70 | 0.41 | 0.24 | 0.22 | 0.23 | **0.87** | 0.23 | **0.84** | 0.24 |

except for grey-fronted dove and tayra *(Eira barbara)*. Distance to conucos had only strong support for tayra, lowland paca and black curassow, moderate support for two species and

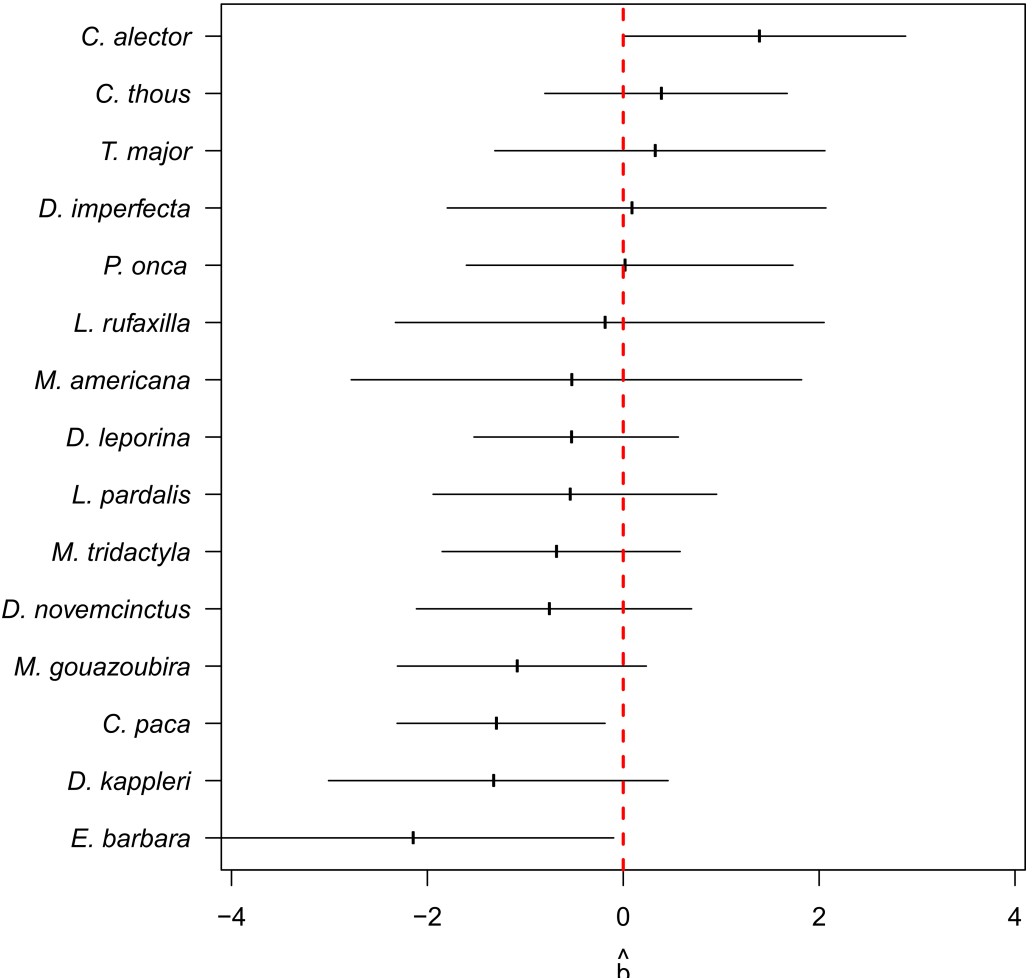

**Figure 4 Conditional RN-model averages of the coefficient of distance to conuco.** Error bars are 95% confidence intervals.

low support for the rest (Table 3). While the distance to the river variable had high values for giant anteater and jaguar, and low for the rest of the species.

Conditional model averages of the coefficient of distance to conuco was negative for most species (higher estimates of latent abundance near to conucos) and close to zero or positive for Guianan white-eared opossum (*Didelphis imperfecta*), great tinamou, crab-eating fox (*Cerdocyon thous*), and black curassow (Fig. 4). However, the 95% confidence intervals of the estimates overlap with zero, except for lowland paca, tayra, and black curassow.

In general, and despite few outliers, abundance predictions from the mixture models were higher for most species in sites where the Pemón reported hunting activity (Fig. 5). This was true for species with different values of hunting preference (*Hv*) and for species not mentioned as important prey for Pemón, including carnivores (with the exception of crab-eating fox; Fig. 5).

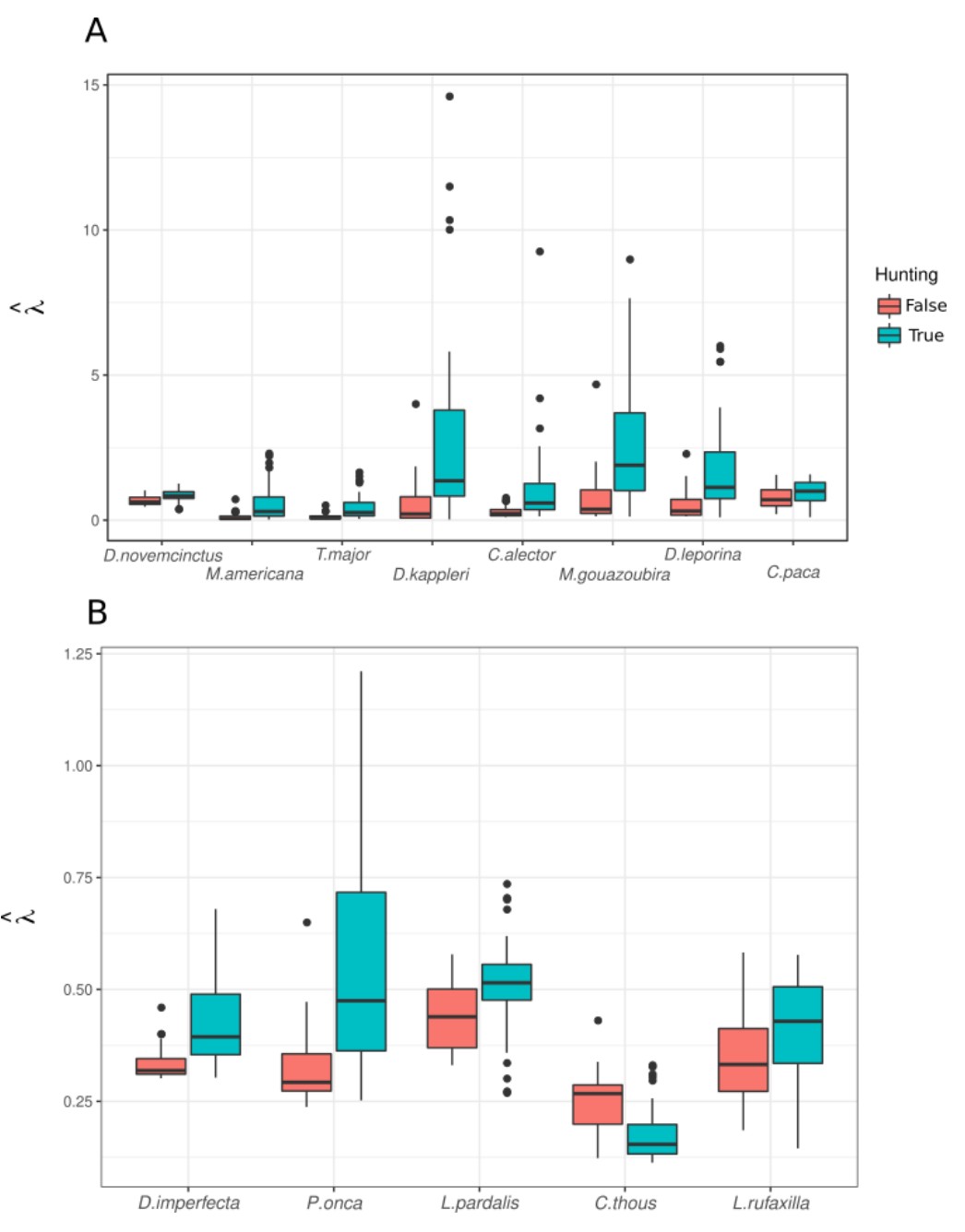

**Figure 5  Predicted abundance over sampling units with and without hunting.** (A) Game species reported by interviewees, ordered from left to right by decreasing Hv value (see Table 2). (B) species not reported as game by interviewees.

## Selection of hunting localities

Hunting occurrence in the study area was detected up to 5 km distance from conucos, both in the savanna and forest (Fig. 2, Supplemental Information 1). Sampling units with reported hunting activity were mostly located at 2.5 km from nearest conucos with tree

**Table 4  Regression coefficients (±SE) and standardized regression coefficient (z value) of each variable explaining hunting occurrence in the Gran Sabana.**

|  | Estimate | Standard error | z value | p |
|---|---|---|---|---|
| Intercept | −1.475 | 0.780 | −1.890 | 0.05877 |
| tree_buffer | 0.041 | 0.015 | 2.726 | 0.00641** |
| dist_river | 0.001 | 0.000 | 1.812 | 0.06998 |
| dist_conuco | 0.000 | 0.000 | −1.276 | 0.202 |

Null deviance: 82.108
Residual deviance: 62.604
AIC: 70.604

Notes.
**: significant term with p < 0.01.

**Table 5  The contingency tables showing preferences for habitat and hunting season among interviewees from the four Pemón communities.**

|  | Habitat | | | Season | | |
|---|---|---|---|---|---|---|
| Community | Forest | Savanna | Mixed | All year | Rainy season | No preference |
| Kami | 4 | 1 | 2 | 1 | 2 | 1 |
| Mare Paru | 5 | 0 | 0 | 1 | 5 | 0 |
| Uroy-Uaray | 4 | 5 | 4 | 3 | 2 | 2 |
| Wuarapata | 8 | 3 | 4 | 1 | 10 | 0 |
|  | X2 = 7.6729 | df = 6 | p ≤ 0.263 | X2 = 9.8886 | df = 6 | p ≤ 0.129 |

cover > 40% (Fig. 2). Tree cover ($p = 0.006$) and distance to rivers ($p = 0.070$) had a positive significant effect on the hunting occurrence, but the effect of distance to conuco ($p = 0.202$) was negative and not significant (Table 4).

Forest was the preferred hunting area for the majority of interviewees (72%), followed by savanna (31%), and mixed forest –savanna areas (34%). This pattern was similar across communities ($\chi^2 = 7.67$; degree freedom = 6; p ≤ 0.263; Table 5). The majority of interviewees hunt during the rainy season (68%), between May and August, while only 21% interviewees hunt all year round, and 11% had not a preferred season to hunt. This pattern was similar across communities ($\chi^2 = 9.89$; degree freedom = 6; p ≤ 0.129) (Table 5).

## DISCUSSION

A clear understanding of the relationship between human activities and wildlife abundance patterns is crucial to identify the most appropriate conservation interventions in complex landscapes with high biological and cultural diversity (*Weinbaum et al., 2013*; *Gavin et al., 2015*). For the Gran Sabana and the Canaima National Park, despite their importance as UNESCO World Heritage Site and the longstanding presence of Pemón people, base-line knowledge about wildlife abundance patterns and how it changes across time, space, and as response to human-based stressors is limited. Our research goes a step forward to fulfill these gaps by providing the first attempt of a systematic sampling survey in the Gran Sabana, generating a quantitative dataset that not only describes the current pattern of

wildlife abundance, but sets the base-line to monitor temporal and spatial changes. Further, to our knowledge, this study is the first in providing quantitative and updated information about Pemón's hunting occurrence, scope and practice, which has been poorly described across the highland Amazon region. Last, but not less important, our hypothesis-based approach allowed us to go beyond a list of species present and hunted, to try to shed light upon underlying patterns that can be better integrated in programs for sustainable use of wildlife in accordance with the cultural and social context. Finally, we place our research in the current social and economic situation of mining encroachment in Guyana Shield.

## Current pattern of wildlife abundance

The vegetation type had the most significant role explaining the abundance pattern of both herbivores and carnivorous species in the study area (Table 3; *Stachowicz et al., 2020*). Most species modeled, except the crab-eating fox, were more abundant in areas with higher cover trees, which may correspond with forest and shrublands (Table 3). In the Gran Sabana, in contrast to other Amazon areas, the savanna ecosystem is more prevalent than forest (*Rull et al., 2013*) thus, the observed higher abundance in forest and shrublands may reflect the patchy distribution of resources (water, shelter and food) in the study area. Although hydric resources had low importance for most modelled species, their inclusion significantly improved Royle-Nichols model performance. In general, neotropical mammals shown higher richness in areas close to water (*Di Bitetti et al., 2008*; *Ferreguetti, Tomas & Bergallo, 2017*). For example, rivers and streams were important to explain abundance pattern for the jaguar (Table 3; *Cullen et al., 2013*), that used to find their preys close to rivers (*Weckel, Giuliano & Silver, 2006*). Also, for the giant anteater, gallery forests along river banks, provides refuge or escape routes from fires (*Diniz & Brito, 2013*).

As expected, the medium and small species, with fast growing rates like the lowland paca and the red-rumped agouti, had the highest frequency of detection (Table 1). The gray brocket, although less frequent, was yet a prevalent species in the area, which contrast with the almost absence of other deer species, the white-tailed deer (Table 1). Formerly widely distributed and abundant, the white-tailed deer was only detected four times across the six survey months (Table 1). This low detection rate was unexpected because this species has a higher tolerance and adaptation capacity to different habitats, than other ungulates such as peccaries and tapirs, being detected even near human population centers (*Gallina & Lopez Arevalo, 2016*). Also, Pemón refers that 10 years ago, the white-tailed deer used to be more abundant in the area (information provided by interviewees). Although currently classified as Least Concern in the national assessment of the Red List of Species (*Rodríguez, Garcia-Rawlins & Rojas-Suárez, 2015*), the current low abundance of the white-tailed deer raised concern about their conservation status, and highlight the need for an in-depth population and threats assessment in this area. Similarly, the other threatened species detected (six Vulnerables and one Nearly Threatened) had also low abundances. Among them, the black curassow and the great tinamou were important for Pemón hunters, generating also concern about the population status in the future (*Rios et al., 2020*).

Evidence supporting the Garden Hunting prediction about higher abundance of small and medium wildlife species close to conucos was not conclusive: Although most of the herbivorous species modeled seem to have higher abundance close to conucos (Fig. 4) this effect was not significant, and the only two species significantly attracted by conucos were tayra and lowland paca. This pattern seems to agree with previous results on which tayra does not show a marked preference for any type of habitat, while lowland paca showed significant preference to shrublands or intermediate habitat, which likely correspond to conuco (*Stachowicz et al., 2020*).

Among the species not attracted by conucos, black curassow was the only showing a significant effect (Fig. 4), which contrasts with previous evidence in lowland Amazon and Piaroa communities where cracids were observed within conucos in high abundance (*Zent, 1997*). These difference might be due to different relationship between indigenous communities and this bird species (with Piaroa using this species as a pet, while for Pemon is a game species), or different habitat preferences of the species between lowland and highland Amazon. For other species of curassow, the Endangered red-billed curassow (*Crax blumenbachii*), in Brazil was more persistent in forest patches faraway from settlements, with hunting pressure potentially exerting more influence on population persistence than habitat quality (*Rios et al., 2020*). Again, more detailed population studies are required to better describe population status of this and other endangered species in Gran Sabana, as well as improve our understanding of landscape transformation and human activities in their population dynamics (*BirdLife International, 2016*).

## Pemón's hunting occurrence and practice

We did not find support for the predictions of higher occurrence of hunting close to conucos (Table 4). Pemón seem to hunt in locations dominated by forest (Table 4 and 5), where species abundance was predicted to be higher (Fig. 5), than in close vicinity of conucos. To our knowledge, there are no studies describing the size of Pemón hunting territories. Here, we found that hunting activity was mostly focused on a radius of 2.5 km from conuco, but we have limited data to test long range hunting (>5 km) (Fig. 2). Evidence from other Pemón community, Tuauken located at ∼ 30 km from study area, describes three types of hunting trips (*Urbina, 1979*): (1) hunting of large mammals such as tapirs and deer, implemented in a planned manner by teams of many people, (2) hunting of smaller animals such as paca or aguti, implemented in a semi-planned way by small teams and even by a single person, and (3) the informal, unplanned hunts of turkeys and birds. The hunting detected in our research likely corresponds with the second and third hunting types: small mammals and birds hunted in hunting trips performed within short to medium distance from conuco.

Traps, incidental capture of game, and even the use of fire to capture deer and other large prey species have been previously reported as hunting methods in Pemón communities (*Bilbao, Leal & Méndez, 2010*; *Sletto & Rodriguez, 2013*). However, during the 120 days of field work, we did not detect traps, supporting the notion that traps are infrequently used by South American indigenous communities (*Dunn & Smith, 2011*). The use of dogs was registered on camera traps but not mentioned by hunters during the interview. Literature

indicates the presence of dogs in hunting zones adjacent to the community, but hunters reported to kills the prey with a firearm, not a dog (*Dunn & Smith, 2011*). We compiled anecdotal information about incidental hunting, mostly pigeons (grey-fronted dove), which were abundant in the study area (Table 1; Fig. 5). However, this activity was not even considered as hunting by interviewees, who dismissed it as part of child games and training.

Hunting scope focused on the most abundant and easily accessible species, the lowland paca, which was the most important hunting prey for Pemón people (Table 2). In lowland Venezuelan Amazon, hunting scope of Ye'kwana and Sanema indigenous appears to be similar to those observed by Pemón communities in this study: they hunt the most abundant mammals in the area (the white-lipped peccary and the lowland paca; *Castellanos, 2001*; *Ferrer et al., 2013*). In a broader geographical context, this focus on high abundant and accessible species (usually pacas, deers and peccaries) was also reported in indigenous communities in Panama (*Smith, 2005*), Honduras (*Dunn & Smith, 2011*), in the Peruvian Amazon (*Francesconi et al., 2018*), French Guiana (*Richard-Hansen et al., 2019*) and in Guiana (*Roopsind et al., 2017*).

The fact that other less abundant or accessible species like the white-tailed deer and the black curassow were identified as important prey for the Pemón, suggest that they practiced selective hunting (Table 2). For the lowland paca with a high reproduction rate and short gestation period (*Grzimek, 2003a*), this selective hunting may not translate into abundance reduction. Indeed, this species has been considered as manioc and maize crop pest in Western Brazilian Amazonia and Honduras (*Abrahams, Peres & Costa, 2018*). Nevertheless, in Ecuadorian Amazon, abundance of lowland pacas, red brockets and collared peccaries have been substantially reduced within a 3 km radius of the communities (*Zapata-Ríos, Urgilés & Suárez, 2009*) and 2 km in western Panama (*Smith, 2008*), raising concern about sustainability of hunting. For the white-tailed deer, with lower reproductive rate and longer gestation period (*Grzimek, 2003b*), this pressure likely had reduced their abundance: frequently hunted in the last decade as source of animal protein and sport hunting (*Danields, 1991*; *Gallina & Lopez Arevalo, 2016*), currently was scarcely reported as hunted. Our current dataset does not allow us to discriminate whether the apparent reduction in abundance of the white-tailed deer is driven by overhunting, demography, environmental or genetics factors (*Madhusudan & Karanth, 2002*; *Grzimek, 2003b*). A sampling design surveying both locations with and without deer hunting across different seasons, and taking into account spatial distribution of potential stressors, will help understand the relative importance of hunting and other factors into deer abundance.

Interestingly, we did not detect reports of human – carnivore conflicts, even though ocelot and tayra were detected close to conucos (Fig. 4). Abundance of ocelot and jaguar were predicted to be higher in Pemón's hunting locations (Fig. 5) suggesting potential competition for prey resources between carnivorous and human. Our failure to find evidence of carnivore poaching or conflict in Pemón communities may be a real pattern and not under-reporting because: (1) in the Pemón communities people openly report hunting for other threatened species such black curassows, (2) in other regions of Venezuela (even very close like Imataca), when a poaching or conflict event exists, people freely exhibit

felids skins at their houses as trophy hunts and talk about the chasing of conflicting animals (IS, pers. obs.), and (3) the cultural taboo in Pemón people regarding hunting of carnivores (*Coppens & Perera, 2008*) seems to be reinforced by more recently adopted religious which restrict the hunting scope only to deers (*Bonet, 2020*; *Knoop et al., 2020*). However, under-reporting is still expected because of poor recall capacity of interviewees. Clearly this topic requires a more in depth research, using specialized questioning techniques widely applied to assess illegal wildlife trade and support sensitive data collection (*Nuno & St. John, 2014*).

## The Garden Hunting hypothesis: current pattern of natural resource use

Our results suggest that social-cultural context, and not only the surrounding environment, determine where and what is hunted. Species attracted by the 'garden' such as lowland pacas, red-rumped agoutis and South American tapirs, were accessible and preferred prey (Fig. 4). In contrast, other species similarly attracted to conucos such as the long-nosed armadillo and the grey-fronted dove, were not preferred as hunting game (Table 2). This result contrasts with the general notion that garden farmers often rely on game hunted in swidden gardens as a key source of protein (*Naughton-Treves, 2002*), but agrees with more nuanced studies evaluating the use and perception of wildlife in local communities in Peru, where 51% of interviewee considered that the wildlife attracted to swiddens gardens bring no benefits for them (*Naughton-Treves, 2002*).

Pemón people traditionally had a very diffusely distributed population (*Coppens & Perera, 2008*), with small and low densely populated settlements around which conucos were cut in mature forest fragments adjacent to open savanna. In the last 30 years, Pemón communities have become more permanent and bigger (*Rull et al., 2013*), resulting in a more intensive land use (shorter than 5–26 years fallow periods that allows forest recovery; *Kingsbury, 2001*), and changes in the conuco locations. Again, there is no evidence of how settlement size could affect hunting practices in Pemón communities. Several authors discuss how settlement nucleation and sedentism around missions, along highways or tourist attractions has led to localized game depletion in the Gran Sabana (*Huber & Zent, 1985*), but without supporting data. Studies from other indigenous communities, the Piaroa, an indigenous group inhabiting in the forested mountains of the Middle Orinoco, that were living in small, scattered, and highly mobile communities until recently (*Mansutti, 1990*; *Zent, 1992*), suggest that increase in the hunting size territory is not proportional to the increase in the population size (*Freire, 2007*). However, Piaroa territories are bigger than those surveyed in the present study, and a study covering a bigger area and more communities is necessary to evaluate the effect of settlement size in hunting practices.

In general, Pemón practice seems to be sustainable but the perceived reduction in abundance of important game species raises concerns for both livelihood sustainability and biodiversity conservation. The general low impact of the current hunting pattern in the Gran Sabana could be explained by their particular economic and cultural context. In the last decade frequency and amount of hunting has been limited because of the high prices of cartridges. The lack of ammunition forced the adoption of traditional, less effective hunting techniques such as bow –arrows and sling, which only allow hunting for small

prey and birds (Figs. 3C, 3D). As a result, hunting has become more incidental, carried out only in special festivities. However, even with cartridges, Pemón people seem to have relied more on conucos' production and fishing as sources of protein, while hunting was a secondary source of protein (*Urbina, 1979*). Additionally, protestant missionaries that have been present for more than a decade in the study area, encourage indigenous communities to vegetarianism and quit hunting.

The new concern regards to creation of a large scale (12,000 km$^2$) and extensive mining development plan the Orinoco Mining Arc in 2016 (OMA; *Lozada, 2019*) in South of Venezuela, which might change the current pattern of managing natural resources in the Gran Sabana. It stands in non-compliance of environmental and indigenous social rights, increasing the risk of pollution, and social and political conflict (*Giordano et al., 2018*) which likely could increase demand for natural resources, including deforestation and over-hunting (*Rodríguez, 2000*). Already, in one of the studied communities, Uroy –Uaray, Pemón people have extracted poor quality gold until 2012, and currently young men are leaving the community to work in legal and illegal mines inside and outside of the Canaima National Park (*SOS-Orinoco, 2018*).

## Study limitations

Carrying out field works in conflict zones such as Venezuela (*Bull & Rosales, 2020*), requires overcoming logistical challenges such as limitation of food and gasoline supply, distrust from local communities, and constant presence of army and paramilitary, altogether impacting safety of researchers and jeopardizing the time and geographical extent of the surveys (*Gaynor et al., 2016*). This challenging social context, combined with budget limitations resulted in short sampling effort, which was limited to six months in the dry season and limited number of cameras. This likely impacted the statistical power of the analysis and limited our ability to detect significant effects (*Kery & Royle, 2015*). Although with this effort we were able to detect 82% of expected mammals species in the study area (*Huber, Febres & Arnal, 2001*; *Stachowicz et al., 2020*), we failed to capture seasonal variations in abundance and occurrence of herbivores and carnivores. For example, collared peccary and white-lipped peccary, were poorly detected during the survey, likely because they perform seasonal movements during the dry season (*Keuroghlian, Eaton & Longland, 2004*).

Although our sampling design optimized spatial coverage, we did not have enough records (37% of species) to fit all species occupancy models. We found an important effect of sampling effort on detectability of species (Table 3), but date of sampling was important only for a few species. Pemón reported that the rainy season was their prefered season for hunting (Table 3), but our sampling survey covered only the dry season, thus we cannot compare how wildlife abundance patterns change across the year (*Ahumada, Hurtado & Lizcano, 2013*). Increasing sampling effort in both temporal and spatial scale, would allow us to get a better picture of their dynamics and variability.

Although our interview sampling size was low, it represented 10% of the total population size and was representative in terms of age distribution. However, low participation in interviews among indigenous groups in Amazon is frequently reported (*Knoop et al., 2020*).

In any case, we are confident that concealment of hunting scope was low: People openly share hunting reports for both threatened and not threatened species. This low level of concealment is likely related to missing law enforcement protocols or tools to evaluate trends and magnitude of wildlife use.

We were able to obtain spatial information of hunting activity in the study area, but a longer presence of at least one year in the study area might assure higher interview success. Extended survey time, combined with daily interviews approached, likely will result in a more accurate and detailed description of hunting patterns (*Jones et al., 2008*), including quantities of prey and frequency of hunting.

## CONCLUSIONS

Large scale analysis of hunting rates might overlook the factors operating locally, such as landscape type and matrix, wildlife diversity, cultural hunting taboos, religion, type of protein sources (fishing, hunting), hunting technology, economic context or emerging threats, leading to misinterpretations and incorrect management decisions. Understanding the relationship between human activities and wildlife diversity patterns is crucial to identify the most appropriate conservation interventions in complex landscapes with high biological and cultural diversity (*Weinbaum et al., 2013*; *Gavin et al., 2015*; *Rovero et al., 2020*).

Our study provides a baseline to evaluate the impact of the growing and accelerated threats in the Gran Sabana ecosystem of highland Amazon. On one hand, the current level of shifting cultivation practices seems to be sustainable and gives a room for sustainable agricultural production in the long term. On the other, cultural transformations, migration of non-indigenous groups for mining activity may possibly generate higher hunting activity. Update of the obsolete legal framework and increased capacity for law enforcement regulating wildlife use will be necessary to avoid local depletion of threatened and preferred prey species. Cost-efficient monitoring strategies will be required to assess the effectiveness of the proposed conservation actions. At regional scale, abundance of functional groups (*Vetter et al., 2011*; *Mason & Mouillot, 2013*; *Rovero et al., 2020*) may be used as an indicator of ecosystem functionality (*Ferrer-Paris et al., 2019*), while at the local scale the occurrence estimates provided by this and similar studies (*Stachowicz et al., 2020*) can be used to calculate maximum sustainable offtake quantitatively combine the supply and demand for wildlife resources (*Robinson & Bennett, 2004*).

## ACKNOWLEDGEMENTS

We especially thank the Pemón communities Kawi, Uroy-Uaray and Wuarapata in the Gran Sabana for supporting this project. Especially thanks to Nigel Noriega for language editing.

### Funding

This work was supported by Idea Wild foundation and Finca Dos Aguas from Venezuela. The funders had no role in study design, data collection and analysis, decision to publish, or preparation of the manuscript.

### Grant Disclosures

The following grant information was disclosed by the authors:
Idea Wild foundation and Finca Dos Aguas from Venezuela.

### Competing Interests

The authors declare there are no competing interests.

### Author Contributions

- Izabela Stachowicz conceived and designed the experiments, performed the experiments, analyzed the data, prepared figures and/or tables, authored or reviewed drafts of the paper, and approved the final draft.
- José R Ferrer-Paris conceived and designed the experiments, analyzed the data, prepared figures and/or tables, authored or reviewed drafts of the paper, and approved the final draft.
- Ada Sanchez-Mercado performed the experiments, analyzed the data, prepared figures and/or tables, authored or reviewed drafts of the paper, and approved the final draft.

### Human Ethics

The following information was supplied relating to ethical approvals (i.e., approving body and any reference numbers):

The study received permits from Instituto Nacional de Parques (INPARQUES) 18/16 205, 156, 17 in Venezuela and local authority in Gran Sabana that authorize human participation in surveys.

Additionally, local indigenous authorities from sector 5 in Gran Sabana gave authorization to carry out the project that includes camera trapping and interviews.

### Data Availability

Code is available at GitHub: https://github.com/icorei/Human_activityGS.

### Supplemental Information

Supplemental information for this article can be found online at http://dx.doi.org/10.7717/peerj.11612#supplemental-information.

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
