# Peer review of "Shifting cultivation and hunting across the savanna-forest mosaic in the Gran Sabana, Venezuela: facing changes"

_PeerJ, doi:10.7717/peerj.11612_

## Round 0.1 · original submission · Major Revisions

After contacting nearly 20 potential reviewers I finally received two very detailed reviews for your paper. Both believe your work has merit but highlight a number of major issues that require a substantial revision. Please address each one of the raised issues, paying special attention to the identified methodological problems (such as reporting goodness of fit of your models, accounting for over dispersion, clarifying your model selection protocol, specifying how you separated detection vs occupancy models, and excluding results with low number of observations). Additionally, please clarify your study design (sample sizes, location of camera traps, etc), and restructure you main study questions as suggested.

·

Basic reporting

The manuscript provides novel insights about the relationship between wildlife and human activities at the Northern amazon based on the Garden hunting hypothesis, despite some caveats (see above), especially in methods sections. A main contribution of the manuscript is a potential baseline to current threats as authors pointed out. In my opinion, it is a valuable contribution for PeerJ. However, it needs some improvement to be considered for publication. With these issues conveniently addressed in a new version, I think that it would be an exciting contribution to this interesting topic for ecology and conservation.

Experimental design

Although authors provide information about the study design, some gaps need to be resolved. For example, how authors assessed the effect of natural heterogeneity of the matrix forest-savanna vs. conocos in modeling? It appears to be considered in the buffer area (1 km). Still, because of the potential effects of the scale, it would be useful to understand what the reasoning was to use 1 km. So, a potential effect of scale needs to be assessed (or provide support to this scale). In some cases, specie-specific effects are scale-dependent. How many camera traps in forest? Savanna? Conucos?

Moreover, the matrix of the study area contrast with the lowland Amazon (Savanna is absent in the Amazon forest), so that this issue needs to be considered in discussions.

Authors omitted a goodness-of-fit (GoF), which is a common recommendation to check the fit of the models (including c-hat). Another option is a plot of residuals (See Kery et al. 2016, Mackenzie et al. 2018, MacKenzie and Bailey, 2004). In fact, due to the low number of detections (for several species) overdispersion need to be assessed by each model. Kery et al. 2016 provide some recommendations to deal with this issue.

They need to include this, in addition to AICc. Indeed, based on the number of detections authors need to use AICc, which is recommended for small samples (Burnham and Anderson, 2002). I suspect that they used ∆AICc < 2, according to Burnham & Anderson et al. (2002). Still, this issue is missing in the manuscript and should be included. It is a standard protocol for modeling.

Why blocks are not considered in modeling? It is possible to assess each block as a potential covariate for detectability. Figure 1 suggests that authors need to include each block as a potential covariate (as a factor) at least for detectability because each block has a different landscape configuration in terms of both forest and savanna. I suggest using Royle-Nichols models, which is recommend to camera trap data (see Tobler et al. 2009, Tobler et al. 2015) because of a relationship between abundance and detectability. For example, in the abstract and discussion authors pointed out the importance of forest cover on mammal diversity; however, they did not include savanna, which is the other primary vegetation type throughout the study area. In fact, authors highlight the importance of savannah for some species (e.g. white-tailed deer)

How many models were included in detection and occupancy by each species? As a common rule, detection models are assessed first. Then occupancy models (Kery et al. 2016) Make it clear for the readers.

There is an effect of a low number of observations (site detections) on occupancy modeling (see standard errors > 1). In these cases, the number of detections is not enough to obtain reliable estimates of the occupancy, so that they are poorly informative about the value of the parameters. For that reason, these should not be considered in results. Provide only results from species with enough data based on standard errors (see Ahumada et al. 2013).

Validity of the findings

The main questions of the studies include:
(1) How much do hunting activity, conuco presence and forest cover affect wildlife occurrence?
(2) Do current hunting pattern of preferred prey and hunting habitat, reflect prey availability or cultural factors?
(3) Does current hunting techniques favour an increase in hunting frequency?
Garden hunting practice is described as hunting of wild animals when these enter swiddens and fallows to take advantage of the resources that these provide them (see Naugthon-Treves 2002 and Linares 1976). However, the manuscript pointed out that huntig occurs in the forest, so I have some doubts if the hypothesis is properly formulated. In fact, camera traps in agricultural areas appear to be missed in the sampling design. Authors only considered dist_conuco as covariate explaining the issue about agricultural areas.

Additional comments

Authors need to include references which test the garden hypothesis (e.g. Naughton et al. 2003)
Line 39. It is "prey."
Line 54-58. I suggest removing this paragraph to start the Introduction with the main idea…since Line 59
Line 102. I recommend removing "Independently of Amazon region."
Line 157. It is not clear how many cameras by block. Each block had 25 sampling units of 2 km2, but authors pointed out 30 cameras by block
Line 257-259. Also, it would be informative a global result?
Line 267. A parenthesis is missed, 27%)
Line 297. It is species richness
Line 297. "Low number of conucos and all mammals' diet groups". Authors need to define functional groups in methods
Line 308. The procedure to estimate deviance needs to be described in methods
Line 331. I suggest using functional traits, according to Jones et al. (2009) and Wilman et al. (2014). For example, tayra is considered omnivorous based on diet data. It is vaguely described in methods (Lines 222-223). Please, include references
Line 456-457. Include some references about functional groups as an indicator of ecosystem functionality (e.g. Mason & Mouillot 2013, Tilman 2001).
Table 3. It is necessary to include the number of detections for each species. Nagelkerke's coefficient of determination is not described in methods.
Authors should have included a table showing the description of the sites in terms of vegetation type and the other covariates where they were located. I suggest it as Supplementary material.

Figure 1. I suggest including a map with the Amazon basin to locate the study area in this context. It will be useful for readers of other regions.
As supplementary material, if it is possible, include pictures of the species, it would be informative for readers of other regions.

MacKenzie, D.I., Nichols, J.D., Royle, J.A., Pollock, K.H., Bailey, L.L., Hines, J.E., 2018. Occupancy estimation and modeling: inferring patterns and dynamics of species occurrence, 2nd Edition ed. Academic Press, Boston.
MacKenzie, D.I., Bailey, L.L., 2004. Assessing the fit of site-occupancy models. Journal of Agricultural, Biological, and Environmental Statistics 9, 300-318.
Kéry, M., Royle, J.A., 2016. Applied hierarchical modeling in ecology: Analysis of distribution, abundance and species richness in R and BUGS: Volume 1: Prelude and Static Models. Academic Press, Boston.
Burnham, K.P., Anderson, D.R., 2002. Model selection and multimodal inference: a practical information-theoretic approach. Springer, New York.
Tobler, M.W., Zúñiga, A., Carrillo-Percastegui, S.E., Powell, G.V.N., 2015. Spatiotemporal hierarchical modelling of species richness and occupancy using camera trap data. Journal of Applied Ecology 52, 413-421.
Tobler, M.W., Carrillo-Percastegui, S.E., Powell, G., 2009. Habitat use, activity patterns and use of mineral licks by five species of ungulate in south-eastern Peru. Journal of Tropical Ecology 25, 261-270.
Ahumada, J.A., Hurtado, J., Lizcano, D., 2013. Monitoring the status and trends of tropical forest terrestrial vertebrate communities from camera trap data: a tool for conservation. PLoS ONE 8, e73707.
Jones, K.E., Bielby, J., Cardillo, M., Fritz, S.A., O'Dell, J., Orme, C.D.L., Safi, K., Sechrest, W., Boakes, E.H., Carbone, C., Connolly, C., Cutts, M.J., Foster, J.K., Grenyer, R., Habib, M., Plaster, C.A., Price, S.A., Rigby, E.A., Rist, J., Teacher, A., Bininda-Emonds, O.R.P., Gittleman, J.L., Mace, G.M., Purvis, A., 2009. PanTHERIA: a species-level database of life history, ecology, and geography of extant and recently extinct mammals. Ecology 90, 2648-2648.
Wilman, H., Belmaker, J., Simpson, J., de la Rosa, C., Rivadeneira Marcelo, M., Jetz, W., 2014. EltonTraits 1.0: Species‐level foraging attributes of the world's birds and mammals. Ecology 95, 2027-2027.
Tilman, D., 2001. Functional diversity. Encyclopedia of biodiversity 3, 109-120.

Reviewer 2 ·

Basic reporting

This research, by combining quantitative data and findings from interviews and analyzing the spatial relationships between Indigenous communities, habitat (including anthropogenic habitat) and game species, has the potential to make an important contribution to our knowledge of a significant (and complex) problem. There are very few studies on the spatial ecology of game species in tropical regions where hunting occurs, in part because it is often very difficult to do in remote regions – but it is critically important for managing wildlife. The camera trap data in particular is robust.

However, there are some weaknesses in this paper that should be addressed. One relates to the methodology, in that there is a somewhat “loose” fit between the data used and what is actually happening on the ground. Some of the indicators are somewhat course and/or imprecise, and there is no data on the actual harvest rates of game species, which is a significant limitation (acknowledged briefly on lines 404-405). However, some of these problems can be resolved with the existing data (as discussed below). A second problem relates to the interpretation and analysis of the data and rethinking some of the claims that are made. The literature review is good in some areas, but very limited when it comes to publications on the spatial patterns of hunting and wildlife abundance (see below).

Additional revisions are required to revise the objectives, clarify aspects of the methodology, pay more attention to ethical issues, and correct numerous minor grammatical errors.

Revision of research questions and objectives (lines 123-126 and elsewhere):

The authors state that they “specifically evaluated”
“How much do hunting activity, conuco presence and forest cover affect wildlife occurrence?” It seems to me that the camera-trap survey should provide the data needed to estimate game species abundance, not simply “occurrence” which I believe is a less precise measure (if not appropriate it would be useful to have some exaplanation of why “occurrence” is the primary variable). The question also includes “hunting activity” which is a bit misleading, as this was not measured. As such, I would re-articulate the research question along these lines
“Are there differences in the relative abundance of game species that are attributable to the presence of (i) conuco agriculture and (ii) hunting activity, measured indirectly by game preferences and proximity to Indigenous communities.” This question seems to capture virtually all of the research that is being presented here, and the authors could simply focus on this one with some further elaboration on the question to explain the research design and the rationale for the approach they use. From my perspective, the other questions below don’t belong. Answers to this first question are very valuable on their own and merit publication, but I will comment on the other ones as well.

“Does current hunting pattern of preferred prey and hunting habitat, reflect prey availability or cultural factors?” This research question is a bit misleading, because data were not collected on “hunting patterns”. Also, the question is not clear.

“Does [sic] current hunting techniques favor an increase in hunting frequency?” This was not measured, so should not be included as one of the research questions. There is some data about the hunting technologies that are reported and anecdotal information about the cost of ammunition, for example, but there is no data on the number of game animals captured using different techniques or the differences in the type and amount game captured by hunters using different techniques.

Finally, all references related to mining should be from the description of the research (lines 126-128) and results sections, and the materials & methods section. It is an important consideration, but the authors did not do any original research related to this. It does belong in the introduction and discussion sections as it is relevant for the significance of the research.

Overall, the paper is well organized. There are numerous grammatical errors, however, too numerous to list here. Further editing will be needed.

Other minor revisions:

The use of the term “ecosystem collapse” (line 96 and elsewhere) is not defined and is also overstating the possible outcomes. Something along the lines of “significant change” is more appropriate. The term “disrupt” is a judgmental term. The term “alter” would be more appropriate.

If possible, separate “hunting and fishing” as separate food sources (line 258).

Camera traps detected 43% of species twice (line 293). Should this be twice or more? As indicated elsewhere, it would be useful to have a summary table of sightings.

Table 3 is difficult to interpret. The authors should explain exactly what the values for “estimate” and “SE” are, as well as “detectability”. Also, for L. pardalis, values are given for “hunting” – shouldn’t this be the same as the others? There are other differences for other species as well in terms of what is presented. The table should be reorganized as a single table of rows and columns, instead of what is there now which is several different tables joined together.

Experimental design

Variables used in the analysis

The authors use “species composition” (line 122 and elsewhere) and “species occurrence” in multiple places, which is useful but they are less robust measures compared to abundance. The camera-trap dataset appears to be sufficiently large for calculating relative abundance or even population density estimates. This would provide a more precise variable for analyzing differences between their sites. For an example, as well as additional methodological issues that should be discussed (e.g., variation in detection probabilities associated with sampling design and species behaviour) along with a stronger rationale for the methodology used. For this they could refer to the article:

“A camera trap assessment of the forest mammal community within the transitional savannah‐forest mosaic of the Batéké Plateau National Park, Gabon” by Hedwig et al. (2018) African Journal of Ecology, Volume 56 (4): 777-790.

This article is one of several in a special issue on camera-trap methods. They could also refer to the following to write a more detailed discussion of potential bias and errors related to camera-trap methods and examples of how to estimate game species population densities:

“Camera trap placement and the potential for bias due to trails and other features” by Kolowski et al., (2017) PloS one, Volume 12 (10).

“Drawn out of the shadows: Surveying secretive forest species with camera trap distance sampling” by Bessone et al., (2020) Journal of applied ecology, Volume 57 (5): 963-974.

“Monitoring the Status and Trends of Tropical Forest Terrestrial Vertebrate Communities from Camera Trap Data: A Tool for Conservation” by Ahumada et al., (2013), PloS one, Volume 8(9).

“Abundance and composition of the medium to large-sized mammals in a private area of a REDD+ project in Acre, Brazil” by Botelho et al. (2018) Biota neotropica, Volume 18 (3).

The variable “n_conuco” is based on “the number of conucos reported or observed in a given block” (line 203). This is a fairly crude measure that does not take into account the size of these farms, when more precise measures seem to be available. The authors present a map that includes land cover categories, and used a buffer analysis for forest cover. If this spatial data is accurate, it would be better to make calculations of agricultural areas based on this same approach for a stronger analysis. This data could replace the “n_conuco” variable as well as the “conuco_dist” variable, which again is a course measure. Moreover, any of these quantitative measures are subject to limitations related to their fit with different species. For species with small territories, forest cover within a 1 km buffer might be a good indicator of the available habitat, but for game animals with larger territories, the land cover over larger areas would be more important. Careful consideration the appropriate buffer distance for different species is needed and a discussion of this limitation is needed in the paper.

Another ecological variable that might explain some of the sighting data is proximity to rivers. This is particularly true for the paca, which if I’m not mistaken, prefers to be close to streams. This factor should be considered in the analysis and interpretation of findings.
The variable “dist_com” might be appropriate, but again seems fairly coarse and requires some justification. For example, a site that is located in between 2 or 3 communities might experience more pressure than another one that is only accessible by one community, even if it is closer. Moreover, there is a large variation in the size of the communities in the study area that likely affects hunting pressure. Uroy-Uaray is about three times larger than Kawi, for example. There are tools available in the GIS software the authors use to make better measures of “closeness” to human settlements, that take into account multiple villages (if appropriate) and differences in population size.
For the camera-trap data, birds and reptiles were later removed from the analysis. How many sightings were used in the analysis then? Was it the reported 7,466 events (line 158), or less? It would be useful to have a table with the breakdown of sightings used in the analysis, by species and each of the 6 blocks.

The authors state that they assessed the “most frequently hunted species” based on “importance of hunting” and “hunting preference” (lines 191-192). Firstly, the “importance of hunting” (which might be better stated as “hunting pressure”) is based on a simple list of game species reported by interviewees. This is problematic, given that the importance of species listed was not ranked in terms of frequency captured or as a food source taking into account body mass. Species that come to mind and that are more memorable, such as the black curassow, might not be the ones that are actually hunted more frequently. This affects what can actually be claimed. For example, the authors state that “These were the most frequently hunted species” (line 275). They do not have sufficient evidence to make this claim. They can only state that they are the most commonly listed game species.

Asking hunters about their game preferences is likewise a weak indicator of hunting pressure, compared to data on the type and amount of game that is actually captured. Preference does not equate to actual harvest rates. For example, armadillos may not be preferred but might actually be captured frequently. Alternatively, a species that is wary or difficult to catch (e.g., arboreal or nocturnal species) might be highly preferred, but captured infrequently. Again, this affects the analysis that can be done and what can actually be claimed.

As an additional analysis that might be useful, the authors could compare the estimated abundance of game animals versus animals that are not hunted by the Pemon (according to the interviews), but this would require an additional analysis with its own complications (e.g., how to determine whether a difference in abundance is attributable to hunting as opposed to a differences related to species characteristics, habitat quality, or something else).

Interview data

More information is needed on how were the interview participants selected and how representative the sample is. If it was not a random sample, this should be stated explicitly and discussed.

The authors note that there were no reports of hunting carnivores (lines 367-368), but do not raise the possibility that legal frameworks might lead to negative consequences for people who hunt endangered species. People might be reluctant to reveal what is considered by officials as illegal hunting. This issue should be addressed.

Related to this, there are ethical issues that are not mentioned. In particular, are there any potential risks posed by the research for local communities who rely on hunting? Could the research findings be used in ways that might affect their hunting rights?

There is some good information on hunting activity as practiced by the Pemon, but some important aspects are neglected. Firstly, given the emphasis on garden hunting, it is important to know something about hunting in agricultural areas. Even people who say that they are not hunters will often capture game opportunistically in their farms. The authors refer to “hunting method” (line 265), but this is incorrect as the results refer to hunting technologies. This suggests that the authors only asked about hunting trips, when other methods might be used – for example, the use of traps or opportunistic capture of game while doing farm work.

This leads to another question: do residents of these communities have dogs? Dogs are not included on the list of hunting technologies. If dogs do not play a role in hunting, this would be unusual and should be stated explicitly. There is also the question of how far hunters go to capture game, discussed above.

There are four communities in the study area according to the authors, but there are five on the map provided, although the fifth one is not named. Is this an error? If not, the presence of this additional community should be addressed.

Note -- I do not feel adequately qualified to review the choice of statistical methods for this study, but I do feel confident in my assessment of the utility of the variables used and overall research design. In any case, some of the statistical methods could change if the authors take my advice.

Validity of the findings

Findings

The research aims to answer questions related to the relationships between game species, habitat, and hunting. Several statements about specific relationships are provided, but the overall pattern that emerges from this research is not entirely clear. The section “Diversity and composition” (lines 289-303) provides several very specific details about game species in different blocks, but again, the overall pattern is not discussed. Answers to the more basic questions should be presented clearly. For example, are there broad differences in relative abundance of game species between sites that are related to the presence (or preferably, prevalence as measured by total area within an appropriate buffer) of conuco farms? Which species are more abundant, less abundant, or the same? A simple scatterplot or line graph would be helpful to discern the overall pattern (e.g., % farmland on the Y axis, and sightings on the X axis), as opposed to the specific examples provided.

The same is true for the influence of distance from Pemon communities – is there a broad difference in sightings related to distance from the nearest community (or preferably in relation to an index that takes into account multiple communities and their population)? Are there significant differences for different species that are either more abundant or less abundant near the communities? Given that these two variables (distance to settlements and amount of farmland surrounding the camera trap) are closely inter-related, the analysis could then examine both together in the occupancy model, after they are examined separately. I recognize that looking at dietary groups is part of the garden hunting hypoethesis, but an additional analysis of individual species would be even more informative. Combining multiple species into dietary groups can obscure significant differences within these groups.

The section entitled “Occupancy models” is similar in providing a series of examples, without an explanation of the overall pattern, which could be presented in the form of text and/or as an additional figure. As discussed above, Table 3 presents results of a model combining several variables and is difficult to interpret and could be reformatted as a single table of rows and columns if possible, not as a series of tables for different species. More explanation in the caption would also be helpful, especially for a broader audience that includes conservation management decision-makers.

A table that presents basic summary data of sightings would be very useful. It would be useful to know how many sightings there were for each species (with more than 8 sightings) in each block and all blocks combined.

The interpretation of the research findings would benefit from engagement with additional studies that have examined the spatial patterns of hunting. There are five that are particularly useful, that they should include in their discussion (and that could also be helpful in how to re-analyze their data and in other ways):

“Spatial patterns of primate hunting in riverine communities in Central Amazonia” by Pereira et al. (2019) Oryx, Volume 53 (1): 165-173.

“The Spatial Patterns of Miskitu Hunting in Northeastern Honduras: Lessons for Wildlife Management in Tropical Forests” by Dunn and Smith (2011) Journal of Latin American geography, Volume 10 (1), p.85-108.

“The spatial patterns of indigenous wildlife use in western Panama: Implications for conservation management” by Smith (2008) Biological conservation, Volume 141 (4), p.925-937.

“Dynamics of hunting territories and prey distribution in Amazonian Indigenous Lands” by Constantino (2015) Applied Geography, 56 222-231.

“Spatial tools for modeling the sustainability of subsistence hunting in tropical forests” by Levi et al. (2011) Ecological Applications, Volume 21: 1802-1818.

There are valuable results highlighted in the discussion section, for example the high number of paca sightings in highly modified areas – although as indicated above, it would be useful to know exactly how many sightings there were for each species. Another conclusion relates to the lower number of deer sightings as being the result of overhunting. The arguments would be more persuasive if there was some measure of number of sightings relative to what would be expected in areas where hunting does not occur, if possible. Given the very different sizes and ecologies of different species, one would expect the number of sightings to vary considerably, even in the absence of hunting.

The faunal composition and relative abundance of species is very important from a conservation perspective. The authors examine species richness in their analysis, but do not comment on the absence or lower relative abundance of species that are of greater concern (although they include IUCN status in Table 3). Overall, is there evidence in their findings that certain species are more vulnerable to the loss of habitat converted to conucos and proximity to communities that include hunters? Or conversely, that the presence of Indigenous communities is beneficial for certain species? The answers are perhaps there in the manuscript, but they are not highlighted as part of the overall pattern that was found in this research.

One very important aspect of the spatial patterns of hunting and game species that is not discussed is source-sink dynamics. Firstly, are there any significant areas in the region that do not experience hunting, or very little hunting. Local depletion of game species might still be sustainable if there are adjacent source areas. The map provided suggests that there may be large areas of forest to the east and northeast.

Another related issue is the spatial distribution of human settlements. The authors explain that the Pemon now live in more nucleated settlements, in a less dispersed pattern. Is there any indication that there is now less hunting pressure in more distant areas? Another related consideration that is not discussed is how far Pemon hunters go when they hunt. For example, do they only hunt in areas that are within one day’s walk, or do they also go on overnight trips to more distant areas? This is not an easy question to answer, but it merits some attention. The literature cited above provides guidance on these issues.

The authors state that “the frequency and amount of hunting has been limited because of the high prices of cartridges” (lines 372-373). What is the evidence that they have to make this claim? They should provide evidence or change the wording to acknowledge the uncertainty involved.

The authors state that collared peccaries “were poorly detected” because of seasonal movements. This is true of the white-lipped peccary, but is not as important factor for this species. Other factors may be important, such as habitat, predation, or hunting.

Overall, from my point of view the analysis presented represents a good first step, but a more careful and thoughtful interpretation and discussion is needed.

Additional comments

As stated in another section, this research has the potential to make an important contribution to our knowledge of a significant and complex problem. There are very few studies on the spatial relationships between game species and Indigenous communities in the neotropics -- or anywhere. The camera trap data is very robust, and with some additional work to rethink the analysis and the interpretation of the findings (that takes into account some of the issues raised in this review and that makes use of the literature on the spatial dynamics of hunting), this will be a much stronger paper.

---

## Round 0.2 · Minor Revisions

The manuscript was improved but the Reviewer and I believe you still needs to address a few important issues.

Please address all the issues raised by the Reviewer. I would like to see your discussion substantially shortened and focused to your results. For instance, the section on "Current threats and opportunities in the Gran Sabana" is interesting but completely unrelated with your work.

Also please standardize terms so that regression coefficients can be easily compared (i.e. Table 4). In Figure 1, Forest-Savanna seems to be a land cover map, not a gradient, so provide meaningful legends for classes. In Figure 2 include regression lines. In all ggplots, please remove the gray background (geom_bw).

·

Basic reporting

No comments

Experimental design

No comments

Validity of the findings

No comments

Additional comments

The manuscript revisions were strong, and I understand and appreciate the points authors made in this new version. It is a remarkable contribution to the Garden hunting hypothesis. If it is possible I would suggest reducing the discussion (~ 3000 words), and prioritizing it according to the main findings of the paper.

General comments
Line 172. Six blocks…because it is the first time that it is mentioned, include in parenthesis (see Sampling design and camera trap survey section)

Line 175 five community leaders known to IS….
personal observation? In parentheses?

Line 283-284. For this study, we calculated tree cover and distance to nearest conuco and river as covariates of species abundance…
Lines 347 – 349 … As site covariates we used tree cover percentage around the camera trap (tree_buffer), distance to river (dist_river), and distance to nearest conuco (dist_conuco)…
It is necessary to standardize terms.

Line 363 …definition of “p” is missed. Why 0.25? It is different to the “p” of occupancy modeling.

Lines 405-406 Thirdly, we compared the predicted abundance of wildlife (based on RN-models see next section) in hunting and not hunting sites.
…see next section?

Dietary group definition is misses in Table 1
Table 2. dsit_rios (dist_rios?) It is in Spanish.

Figure 5. Scientific names in cursive

---

## Round 0.3 · accepted · Accept

I am happy to accept your manuscript but have one final minor request. In your abstract conclusions, please specify that your study "sets the base-line to monitor temporal and spatial change in the Gran Sabana ecosystem of highland Amazon." Your findings are unlikely to serve as base-line for the entire Amazon basin, so please be specific. Congratulations on your work.